# Decadal-scale progression of the onset of Dansgaard–Oeschger warming events

Tobias Erhardt[1], Emilie Capron[2, 3], Sune Olander Rasmussen[2], Simon Schüpbach[1], Matthias Bigler[1], Florian Adolphi[1, 4], and Hubertus Fischer[1]

[1]Climate and Environmental Physics, Physics Institute & Oeschger Center for Climate Change Research, University of Bern, Sidlerstrasse 5, 3012 Bern, Switzerland
[2]Physics of Ice, Climate and Earth, Niels Bohr Institute, University of Copenhagen, Tagensvej 16 2200 København N, Denmark
[3]British Antarctic Survey, High Cross, Madingley Road, Cambridge, CB3 0ET, UK
[4]Quaternary Sciences, Department of Geology, Lund University, Sölvegatan 12, 22362 Lund, Sweden

**Correspondence:** Tobias Erhardt (erhardt@climate.unibe.ch)

**Abstract.** During the last glacial period, proxy records throughout the Northern Hemisphere document a succession of rapid millennial-scale warming events, called Dansgaard–Oeschger (DO) events. A range of different mechanisms have been proposed that can produce similar warming in model experiments, however the progression and ultimate trigger of the events is still unknown. Because of their fast nature, the progression is challenging to reconstruct from paleoclimate data due to the limited temporal resolution achievable in many archives and cross-dating uncertainties between records. Here we use new high-resolution multi-proxy records of sea-salt (derived from sea spray and sea ice over the North Atlantic) and terrestrial (derived from the Central Asian deserts) aerosol concentrations over the period 10-60 ka from the Greenland NGRIP and NEEM ice cores in conjunction with local precipitation and temperature proxies from the NGRIP ice core to investigate the progression of environmental changes at the onset of the warming events at annual to multi-annual resolution. Our results show on average a small lead of the changes in both local precipitation and terrestrial dust aerosol concentrations over the change in sea-salt aerosol concentrations and local temperature of approximately one decade. This suggests that, connected to the reinvigoration of the Atlantic Meridional Overturning Circulation and the warming in the North Atlantic, both synoptic and hemispheric atmospheric circulation change at the onset of the DO warming, affecting both the moisture transport to Greenland and the Asian monsoon systems. Taken at face value, this suggests that a collapse of the sea-ice cover may not have been the initial trigger for the DO warming.

## 1 Introduction

Ice-core records from Greenland reveal millennial-scale warming episodes in the course of the last glacial period, called Dansgaard-Oeschger (DO) events (Dansgaard et al., 1993; NGRIP project members, 2004). During their onset, temperatures in Greenland increased rapidly by 10-15 °C from cold stadial (GS, Greenland Stadial) to warmer interstadial (GI, Greenland Interstadial) conditions within a few decades (Kindler et al., 2014; Huber et al., 2006; Severinghaus, 1999), coinciding with an almost doubling of the local snow accumulation (Andersen et al., 2006). Moreover, aerosol records from Greenland ice

cores show coinciding rapid changes in aerosol concentrations during the onset of the events. These changes are partly caused by reduced atmospheric lifetime of the aerosols due to increased precipitation scavenging en-route and partly by changes in aerosol sources (Fischer et al., 2015; Schüpbach et al., 2018).

Other proxy records throughout the Northern Hemisphere also document the widespread environmental imprint of these rapid warming events. Marine sediment cores (Dokken et al., 2013) and aerosol records from Greenland (Spolaor et al., 2016) show a reduction in perennial sea-ice cover in the Nordic Sea and Arctic Basin and ocean circulation proxies indicate an increase in the Atlantic meridional overturning circulation (AMOC) (Lynch-Stieglitz, 2017). Records from North America indicate a change in the moisture advection from the Pacific with dryer conditions during the warm interstadial periods likely related to changes in atmospheric circulation (Wagner et al., 2010; Asmerom et al., 2010). These circulation changes coincide with increased wild-fire activity in North America as clearly imprinted in the Greenland ice-core record (Fischer et al., 2015). Furthermore, records from Eurasia indicate rapid changes in the local ecosystems (Rousseau et al., 2017). In the lower latitudes, speleothem and sediment records from both South America (Wang et al., 2004; Deplazes et al., 2013) and Eastern Asia (Wang et al., 2008) indicate a northward displacement of the Inter Tropical Convergence Zone (ITCZ) at the time of DO warming (Cheng et al., 2012) resulting in rapid changes in tropical hydro-climate and methane emissions from the tropical wetlands (Baumgartner et al., 2014). A recent synchronization of cosmogenic radionuclide records from ice cores and low-latitude speleothems back to 45 ka ago shows that atmospheric circulation changes in the tropics occurred synchronously with the Greenland warming within the cross-dating uncertainties of around 180 yr (Adolphi et al., 2018). The atmospheric circulation changes associated with the Asian monsoon systems documented in the Asian speleothems also reduced the mobilization and export of mineral dust aerosol from the Central Asian deserts during the warm stadial periods as documented by downstream sediment records (Porter and Zhisheng, 1995; Jacobel et al., 2017).

A range of different, non-exclusive mechanisms that produce interstadial-like warming events during preindustrial and glacial climate conditions have been proposed and tested in model experiments. These include the direct modulation of the AMOC by freshwater addition to the North Atlantic (e.g. Knutti et al., 2004), where ceasing artificial freshwater forcing in the North Atlantic results in an increasing strength of the AMOC and subsequently increasing Greenland temperatures. Many of the proposed mechanisms involve the reduction of North Atlantic sea-ice cover either as driving or as amplifying process for the warming. In both coupled and uncoupled model experiments, the removal of winter sea-ice cover in the North Atlantic and Nordic Seas alone generates an increase in Greenland temperatures and snow accumulation rates similar to observations by exposing the relatively warm underlying ocean (Li et al., 2010, 2005). Experiments with coupled atmosphere and ocean show that this reduction in sea-ice cover can be induced by changes in the wind stress over the sea ice either arising spontaneously (Kleppin et al., 2015) or controlled by elevation changes of the Laurentide ice sheet (Zhang et al., 2014). Furthermore, heat accumulation under the sea ice during stadials can destabilize the otherwise strongly stratified water column, eventually leading to breakdown of the stratification and melting of the sea ice from below (Dokken et al., 2013; Jensen et al., 2016). Alternatively, Peltier and Vettoretti (2014) and Vettoretti and Peltier (2015) describe the occurrence of spontaneous DO-like oscillations in their fully coupled model caused by the buildup and break down of a meridional salinity gradient between the open and the sea-ice-covered North Atlantic. The collapse of the salinity gradient then leads to a rapid disintegration of the sea-ice cover

and an invigoration of the AMOC. In summary, it is clear that sea ice is a crucial factor in generating the full climatic change seen at the DO warming (Li and Born, 2019). However, the sequence of events, and whether the sea-ice loss is triggered by atmospheric or oceanic changes, differs between the proposed mechanisms.

In any case, the progression and possible interaction between the distinct processes that can partake in the DO warming are difficult to constrain from paleo-observations and their validity therefore difficult to test, especially as some of the proposed mechanisms lead to virtually indistinguishable model results (Brown and Galbraith, 2016). The relative timing between the start of DO events in proxy records of different parts of the Earth system can yield critical insight into their spatio-temporal progression and possible causal relations. However, due to the fast onset of the DO events, relative time differences are expected to be in the order of years to decades, requiring very high temporal resolution of paleo-records and often unattainable small relative dating uncertainties between records from different archives.

## 2    Data and Methods

Multi-proxy records from ice cores are ideally suited for this type of investigation as they contain information about different parts of the Earth system in the same archive and thus with negligible relative dating uncertainty when measured on samples from the same depths. Two studies have previously tapped into this potential and used high-resolution proxy data from the NGRIP ice core to investigate the onset of the Holocene, GI-1 and GI-8c using two different approaches to infer the timing of changes in the different proxies (Steffensen et al., 2008; Thomas et al., 2009). In the study presented here we greatly expand on these two studies by using new high-resolution records of mineral dust and sea-salt aerosol as indicated by calcium ($Ca^{2+}$) and sodium ($Na^+$) concentrations from both the NGRIP (NGRIP project members, 2004) and NEEM (NEEM community members, 2013; Schüpbach et al., 2018) Greenland deep ice cores. The data sets span the complete time interval from 10 to 60 ka and include all interstadial onsets from GI-17.2 to the onset of the Holocene. The aerosol records were measured using continuous flow analysis, allowing for exact co-registration of the aerosol concentration records at the millimeter scale and resulting in sub- to multi-annual resolution, depending on the thinning of the ice (Röthlisberger et al., 2000; Kaufmann et al., 2008). Sodium concentrations were determined using an absorption photometric method, calcium concentrations using a fluorimetric detection (Sigg et al., 1994; Röthlisberger et al., 2000; Kaufmann et al., 2008). We further use the NGRIP GICC05 annual layer thickness record, based on the identification of seasonal variations in the aforementioned aerosol and visual stratigraphy records, as a measure of relative local accumulation rate changes (Svensson et al., 2006, 2008) as well as 4-7 yr (5 cm) resolution $\delta^{18}$O measurements on the ice from NGRIP as a proxy for local temperature changes (NGRIP project members, 2004; Gkinis et al., 2014). All data are shown in Figure 1 in decadal resolution on their respective time scales. Between the individual datasets, the co-registration uncertainty is limited by the absolute depth assignment of the datasets. This uncertainty is typically on the order of a few millimeter for CFA data and around a centimeter between $\delta^{18}$O and CFA data, which translates to a co-registration uncertainty in the sub-annual range.

For both ice cores, the current versions of the GICC05 age scale were used, which in the case of NEEM has been transferred from NGRIP using volcanic match points (Rasmussen et al., 2006; Andersen et al., 2006; Svensson et al., 2006; Rasmussen

et al., 2013). All ages are given relative to 1950. Even though at the volcanic match points the dating uncertainty is relatively small, the uncertainty introduced by the interpolation between the match points generally precludes direct comparison of absolute timing between the two cores with the precision required for this study.

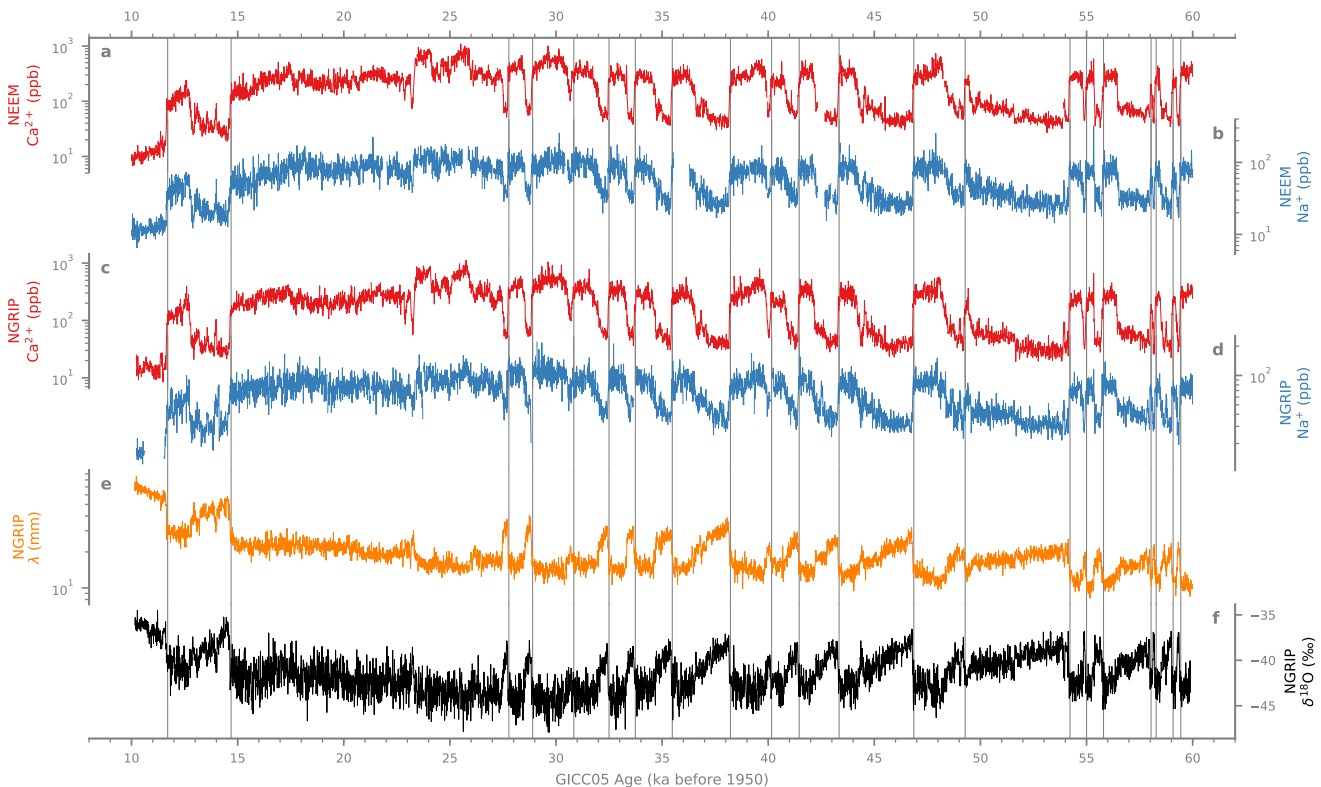

**Figure 1.** Investigated records from the NEEM (a, b) and NGRIP ice cores (c-f). (a, d) show the new aerosol records of $Ca^{2+}$ (a, c) and $Na^+$ (b, d) alongside the NGRIP layer thickness ,$\lambda$ (e) and ice $\delta^{18}O$ records (f). All records are shown as decadal averages on their respective version of the GICC05 age scale relative to 1950 (Svensson et al., 2008; Rasmussen et al., 2013). The vertical lines mark the investigated interstadial onsets as given by Rasmussen et al. (2014). Note that the vertical scales for the aerosols and annual layer thickness are logarithmic.

Mineralogy and isotopic composition of mineral dust aerosol from central Greenland indicate that its dominant glacial
5  sources are the central Asian Taklamakan and Gobi deserts (Biscaye et al., 1997; Svensson et al., 2000). Large dust storms occurring during spring can lift dust up into the westerly jet stream where it is transported over long ranges and at high altitudes (Sun et al., 2001; Roe, 2009). The dust emission and subsequent entrainment into the jet is strongly dependent on the specific synoptic circulation in the area which itself is governed by the latitudinal position of the westerly jet (Nagashima et al., 2011; Roe, 2009). Under current conditions, the position of the westerly jet over central Asia varies seasonally, changing from south to
10  north of the Tibetan plateau during late spring, in unison with the northward movement of the East Asian Summer Monsoon rain

belt (Schiemann et al., 2009; Yihui and Chan, 2005). During the last glacial, the jet was located further south, and, especially during cold periods, enabling more frequent or even permanent conditions for the deflation and entrainment of the central Asian dust (Chiang et al., 2015; Nagashima et al., 2011). Once in the jet stream, the mineral dust aerosol is transported above the cloud level and is largely protected from scavenging by precipitation and transported efficiently to Greenland (Schüpbach et al., 2018). This allows us to interpret changes in calcium concentration records in terms of changes in the conditions needed for dust entrainment, i.e. the latitudinal position of the westerly jet and the Asian hydro-climate.

Sodium-containing sea-salt aerosols are produced either by bubble bursting at the surface of the open ocean or blowing saline snow on the surface of sea-ice (Wagenbach et al., 1998; Yang et al., 2008). Sea-salt aerosol is transported together with moist air masses from the North Atlantic onto the Greenland ice sheet (Hutterli et al., 2006). Under current conditions, aerosol transport models point at the emission from blowing snow from winter sea ice as the dominant source of sea-salt aerosol for Greenland in winter and suggest that this source is responsible for a large fraction of the seasonal variability seen in ice-core records (Huang and Jaeglé, 2017; Rhodes et al., 2017). However, on inter-annual time scales the influence of the atmospheric transport and deposition dominates the variability at the central Greenland sites for recent times, due to the overall low contribution of sea-ice-derived aerosol to the total sea-salt aerosol budget (Rhodes et al., 2018). Under glacial conditions, the extended multi-year sea-ice cover moves both sources of open-ocean and sea-ice-derived sea-salt aerosols further away from the ice-core sites. Intuitively, this would lead to a reduction in sodium deposition on the ice sheet due to the longer transport. However, the opposite is observed in the Greenland ice-core records of the last glacial with much higher concentrations during cold climate periods than during warmer periods (Schüpbach et al., 2018; Fischer et al., 2007). Isolating the effect of the more distal sources, Levine et al. (2014) have shown that the more extensive sea ice around Antarctica under glacial conditions would lead to reduced sea-salt aerosol transport to the ice sheet. This effect, however, is overcompensated by changes in the atmospheric circulation that enhance production and transport during the glacial in comparison to present day conditions, leading to observed increase in sea-salt aerosol concentrations in Antarctic ice cores (Wolff et al., 2010). This suggests that changes in the transport and deposition regimes must be responsible for a large fraction of the glacial/interglacial and stadial/interstadial variability observed in the Greenland ice-core records (Schüpbach et al., 2018; Fischer et al., 2007). One plausible explanation for the apparently more efficient transport of sea-salt aerosol to Greenland during the cold periods are the drier conditions over the cold, sea-ice-covered North Atlantic compared to open-ocean conditions. This implies that changes in the sea-ice cover do affect the sea-salt aerosol concentrations in the Greenland ice cores not only because of an influence on the source of the sea-salt aerosol but also on the efficiency of its transport, i.e. its deposition en-route. Reduced sea-ice cover increases evaporation from the open ocean, resulting in increased scavenging en-route and subsequently reduced transport efficiency of sea-salt aerosol to the Greenland ice sheet, especially because of its co-transport with moist air masses from the North Atlantic (Hutterli et al., 2006). In turn, this allows us to interpret the stadial/interstadial changes in sodium concentrations in the ice cores as qualitative indicators of the extent of the sea-ice cover in the North Atlantic.

In combination, the records of water isotopic composition, annual layer thickness and $Na^+$ and $Ca^{2+}$ concentrations in the ice allow us to study the phasing between changes in local temperature and precipitation on the Greenland Ice Sheet, North Atlantic sea-ice cover and dust deflation from the Central Asian deserts, respectively. To quantify this phase relationship we

employ a probabilistic model of the transitions to determine their individual start and end points as well as the uncertainties of these points. The model describes the stadial-to-interstadial transition as a linear (in case of $\delta^{18}O$) or exponential transition (i.e. a linear transition fitted to the log-transformed data for all other records) between two constant levels, accounting for the intrinsic multi-annual, auto-correlated variability of the proxy records using an AR(1) noise process. This description of the

transition is similar to the one used by Steffensen et al. (2008), who used a bootstrapping parameter grid-search algorithm to determine the ramp parameters and their uncertainties (Mudelsee, 2000). However, here we use probabilistic inference to determine the parameters of the ramp as well as those describing the multi-annual variability of the records and their uncertainties, conditioning on the data. Inference is performed on the model using an ensemble Markov Chain Monte Carlo sampler (Goodman and Weare, 2010; Foreman-Mackey et al., 2013) to obtain posterior samples for all parameters of interest

as well as the parameters of the AR(1) noise model. In this way, any correlations between parameter estimates and their uncertainties are transparently accounted for. More details on the inference and the mathematical description of the probabilistic model can be found in the Appendix. In addition to the start- and end points we estimate the temporal mid points of the transitions, as they are less influenced by the multi-annual variability that dominates the uncertainties of the timing estimates. However, here we focus on the interpretation of the phase relationship of the parameters at the onset of the stadial-to-interstadial

transitions to constrain the causal relationships of the trigger of the DO-events.

The transition model is individually applied to approximately 500 yr sections of the data at the highest constantly available temporal resolution (i.e. number of years per observation) around the stadial to interstadial transitions in each of the records on their respective time scales (Svensson et al., 2006, 2008; Rasmussen et al., 2014). The exact width of the window was adjusted to account for gaps in the data and the onset of other transitions within the 500 yr window. Because of the thinning of layers

due to ice flow, the time resolution decreases from 2 yr for the onset of the Holocene to 3 yr at the onset of GI-17.2 for the NGRIP CFA data, from 2 to 4 yr for the NEEM CFA data and from 4 yr at the onset of the Holocene to 7 yr for the NGRIP isotope data. The annual layer thickness data was down-sampled to match the NGRIP CFA data resolution. Note, that using the overall lowest available temporal resolution (7 yr) for the analysis leads to practically the same results, albeit with slightly larger uncertainties. It is also worth noting that the largest source for uncertainties in our estimates stems from the multi-annual

variability of the proxy records that cannot be alleviated by higher resolution records.

An example of one of the fitted transitions, the onset of GI-8c in $Ca^{2+}$, is shown in Figure 2 alongside marginal posterior distributions for the onset, mid point and end of the transition. The absolute timing estimates for each proxy are used to infer the leads and lags between the different records, propagating all uncertainties. In the following, all estimates from the probabilistic inference are given as marginal posterior medians and their uncertainties as marginal 90% credible intervals.

As pointed out above, the cross-dating uncertainty between the two ice cores does not allow for sufficiently precise inter-core comparison of the absolute timing of the transitions, however the comparison of the lags between aerosol records of one ice core to those of another are not affected by these uncertainties. Under the assumption that the DO-events show the same imprint in both ice-core records, this enables crosschecking of the results between the two records.

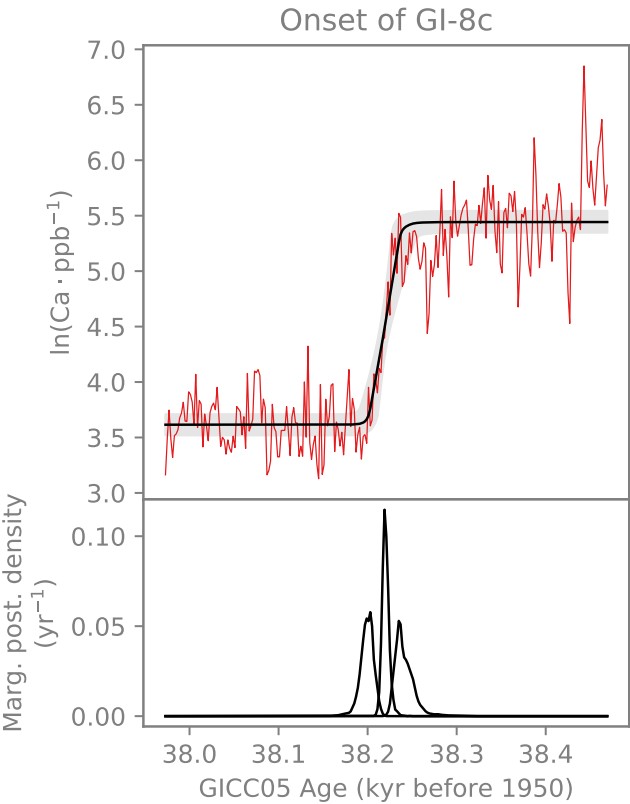

**Figure 2.** Example of a fitted ramp for the $Ca^{2+}$ data from the onset of GI-8c. The top panel shows the data together with the marginal posterior median of the fitted ramp (thick black line). Shaded areas indicate the marginal posterior $5^{th}$ and $95^{th}$ percentiles. The bottom panel shows the marginal posterior densities for the onset, mid point and end point of the transition (from right to left).

## 3 Results

Time differences of the onset, mid points and end points of the stadial-to-interstadial transitions in $Ca^{2+}$, $\delta^{18}O$ and annual layer thickness relative to the transition in $Na^+$ are shown in Figure 3. All inferred change points and timing differences can be found in the Supplementary Data. The timing differences for the $Ca^{2+}$ and layer thickness records relative to $Na^+$
5  for individual events are subtle and their uncertainties are large, but overall they indicate a consistent picture of the phase relationship between the different events, which is the same for the two ice cores. For most events the decrease in $Ca^{2+}$ concentrations lead the decrease in $Na^+$ at the beginning, mid point and end of the transition. Similarly, the beginning of the increase in annual layer thickness slightly leads the start of the decrease in sea-salt aerosol concentrations, though the detected lead is more variable between individual DO events. Nevertheless, the data is overall consistent with a lead of the
10  increase in annual layer thickness at the mid point of the transitions. In the case of the $\delta^{18}O$ record, timing differences are much more variable between the individual events with no clear tendency for neither leads nor lags relative to $Na^+$. For all

of the transitions, the inferred timing differences relative to the onset of the transition in $Na^+$ are smaller than the duration of the transition itself in each of the proxy records. That means that none of the proxies exhibit a complete stadial-interstadial transition before the onset of the transition in the sea-salt aerosol concentration.

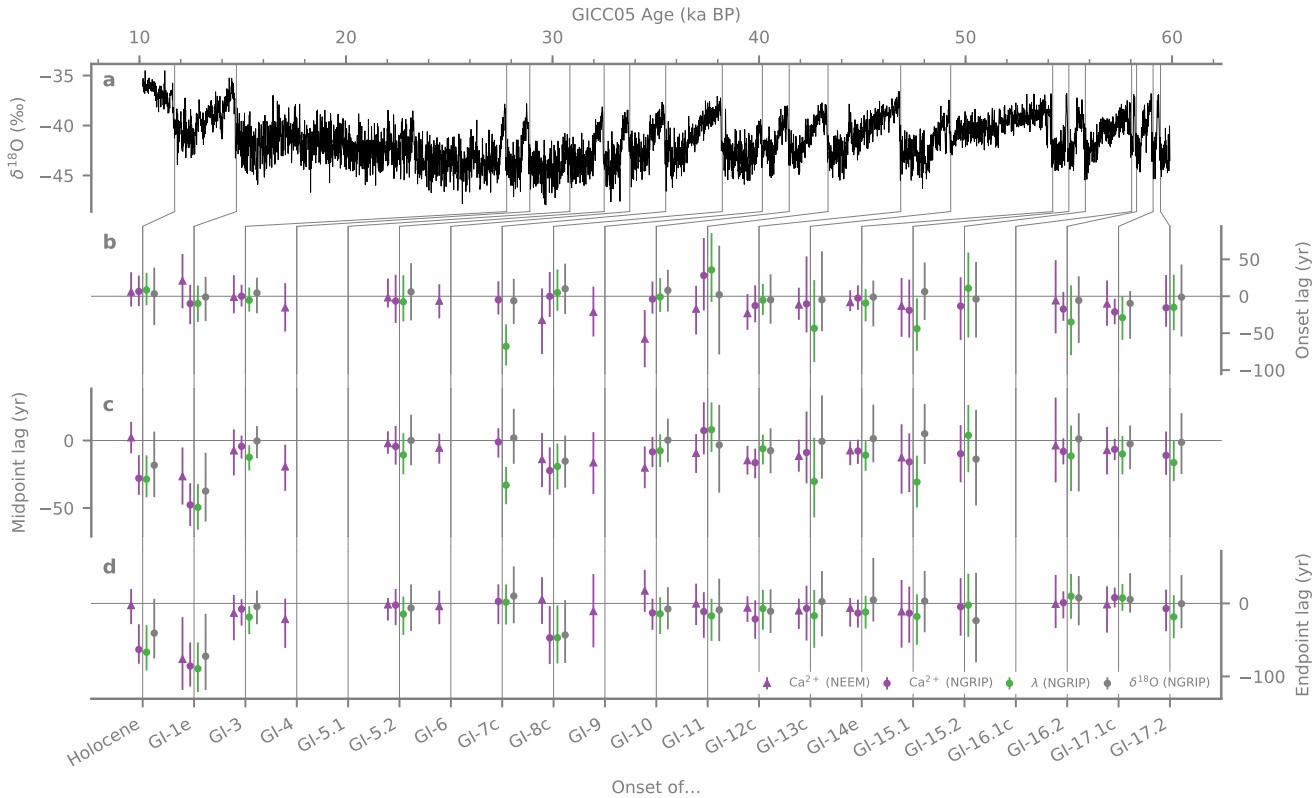

**Figure 3.** Timing differences for the individual interstadial onsets: (a) shows the NGRIP $\delta^{18}O$ record, (b-d) timing difference of NGRIP $Ca^{2+}$, layer thickness ($\lambda$) and $\delta^{18}O$ records as well as the NEEM $Ca^{2+}$ record relative to the transition in the respective $Na^+$ records at the onset (b), mid point (c) and end (d) of the transition. No timing results are given for transitions where there are data gaps in one of the necessary data sets. Error bars show the marginal posterior $5^{th}$ and $95^{th}$ percentiles and the symbol the marginal posterior median. Note the different axis scaling for the start, middle and end points. All inferred absolute timings, the transition durations and the lags relative to $Na^+$ can be found in the Supplementary Material.

For each individual transition the uncertainties are large compared to the leads and lags and only a few events show leads that are bigger than zero with a probability larger than 95 %. However, based on the overall tendency of a lead for both $Ca^{2+}$ and layer thickness relative to $Na^+$ between the GI onsets, we combine the individual estimates of the leads and lags for each core to determine an average timing difference for the investigated proxies. To combine the estimated timing difference of the individual DO onsets, Gaussian kernel density estimates of the posterior samples were multiplied together. Note that this implicitly assumes that the timing differences for all interstadial onsets in the parameters investigated here are the result of

the same underlying process, or in other words, are similar between the interstadial onsets. This assumption differs from the assumption used in other studies of relative phasing of southern and northern hemisphere climate and changes in precipitation source regions during the last glacial (WAIS Divide Project Members, 2015; Markle et al., 2016; Buizert et al., 2018). In these studies, the stacking of the climate events assumes that the complete progression of the climate event is a realization of the

same underlying process which is a wider and more restrictive assumption as it encompasses the whole transition and not only its onset. The combined estimates are calculated for the data from the two cores separately so that the consistency of the timing between $Na^+$ and $Ca^{2+}$ between records can be tested. Probability density estimates for the lead of the other parameters relative to $Na^+$ at the transition onset, mid point and end point are shown in Figure 4. They clearly show that, on average, both the reduction in terrestrial aerosol concentration as well as the increase in annual layer thickness precede the reduction in

sea-salt aerosol for all stages of the transition, whereas no significant lead or lag is identified between $\delta^{18}O$ and $Na^+$.

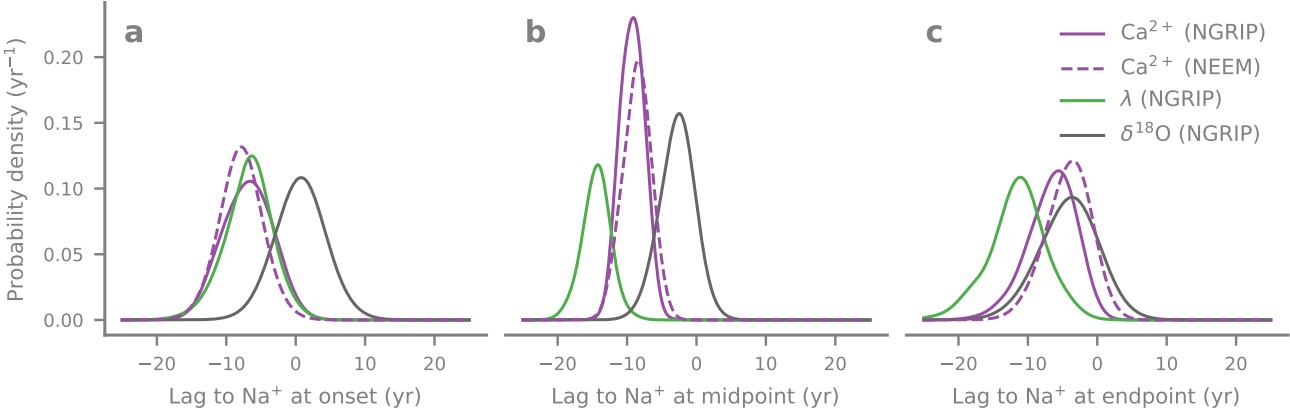

**Figure 4.** Combined probability density estimates of the lag of the $Ca^{2+}$, layer thickness and $\delta^{18}O$ transitions relative to the respective point in the $Na^+$ records for the onset (a), mid point (b) and end (c) of all transitions from stadial to interstadial for the two cores.

In the combined estimate, $Ca^{2+}$ concentrations start to decrease $7^{+6}_{-6}$ yr before $Na^+$ in the NGRIP record and $8^{+5}_{-5}$ yr in the NEEM record, where the error margins refer to the 5[th] and 95[th] marginal posterior percentiles. They reach the mid point of the transition $9^{+3}_{-2}$ yr and $8^{+4}_{-3}$ yr before $Na^+$ and the end point $6^{+6}_{-5}$ yr and $4^{+6}_{-5}$ yr, respectively.

Furthermore, local accumulation rates at NGRIP start to increase $7^{+6}_{-6}$ yr earlier than $Na^+$ starts to decrease, reaching the

mid point $15^{+3}_{-3}$ yr and its interstadial level $12^{+7}_{-6}$ yr earlier.

Note, that the density functions shown in Figure 4 cannot be used to infer timing differences between the other parameters. This is a direct result of the estimates being conditional on the timing of the transition in sodium, leading to large correlations between the lag estimates for the other parameters. That means that even though e.g. two probability density functions of the differences relative to the transition in sodium largely overlap, that does not necessarily mean that their relative timing

difference is equal to zero. In the case of the timing difference between the transition onsets of the increase in annual layer thickness and the decrease in $Ca^{2+}$ concentrations the combined lead of the change in annual layer thickness relative to $Ca^{2+}$

is not larger than zero at the 0.95 probability level with $4^{+4}_{-5}$ years. To establish the most probable sequence of events at the transition offset, we calculate the average order of the onset times, shown for NGRIP in Figure 5. The average positions show that the change in accumulation and $Ca^{2+}$ concentrations about equally likely occur first whereas the transitions in $Na^+$ and $\delta^{18}O$ about equally likely occur last. The same analysis for the NEEM results confirms this sequence.

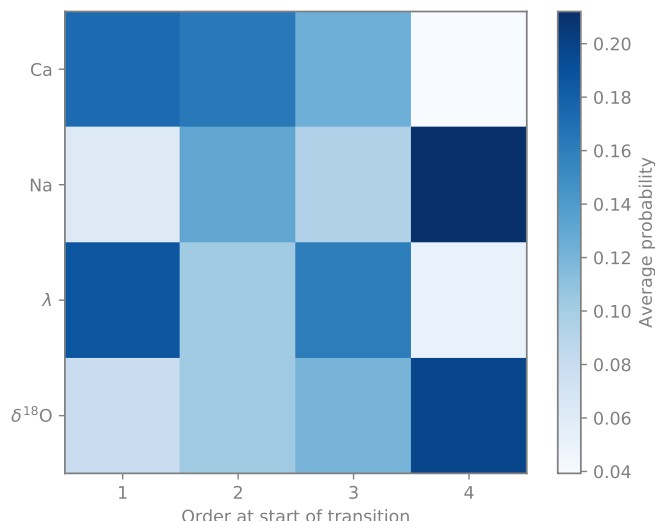

**Figure 5.** Combined order statistics for the onset of the transition. The colors indicate the probability of observing the onset of the transition in the parameter on the y-axis at the position on the x-axis. This illustrates that during the onset of the DO-warming, the changes in accumulation and $Ca^{2+}$ concentrations about equally likely occur first whereas the transitions in $Na^+$ and $\delta^{18}O$ about equally likely occur last.

## 4 Discussion

Using much lower resolution mineral dust and ice $\delta^{18}O$ data from NGRIP, Ruth et al. (2007) reported coinciding onsets within 5 to 10 years and a combined lag of $1 \pm 8$ yr for GI-1 through GI-24 demonstrating the close connection between the Asian and North Atlantic climate during all of the DO events. In light of the low temporal resolution of the data used (0.55 m corresponding to 7–98 yr or 38 yr on average) and the small timing differences that emerge in the combination of the individual interstadial onsets presented here, the results of Ruth et al. (2007) are compatible with the outcomes presented here. Our results are also in good agreement with the more detailed studies of the onset of the Holocene and GI-1 (Steffensen et al., 2008) and GI-8c (Thomas et al., 2009). Similar to the presented study, Steffensen et al. (2008) and Thomas et al. (2009) (using high-resolution data from the NGRIP ice core) also infer slight leads of changes in terrestrial aerosol concentrations and accumulation rate and moisture sources ahead of the changes in marine aerosols and local temperature in good agreement with our results from the respective DO-onsets. These results are also overall in good agreement with the other transitions investigated here. As the

study presented here now covers 19 warming events starting from 60 ka ago and independently investigates the phasing in two ice cores, this adds significant evidence to the initially inferred phasing.

Two studies into Greenland (Schüpbach et al., 2018) and Antarctic (Markle et al., 2018) aerosol records have recently shown that a large part of the variability on centennial and millennial time scales in these records can be explained by the tight coupling between the hydrological cycle and the aerosol transport to the ice sheets. This coupling is a direct result of the efficient removal of aerosols during transport by precipitation scavenging leading to a negative correlation between en-route precipitation and aerosol concentration in the ice. On the short time scales investigated here, this tight coupling breaks down. This has previously been indicated for Antarctica by the lack of coherence at shorter periods (Markle et al., 2018) and here by the lack of synchronous changes between snowfall, water isotopic composition and/or aerosol concentrations during the onset of the warming events. This can be explained by the observation that under glacial conditions, models show that the interannual variability of the accumulation is governed by changes in the frequency of the advection of marine air masses to the Greenland ice sheet (Pausata et al., 2009; Merz et al., 2013). Furthermore, the overall reduced amount of snowfall during the glacial is a result of generally decreased moisture advection (Merz et al., 2013). For the stadial/interstadial transitions, these model results and the lack of synchronous changes in the data presented here suggest a large influence of atmospheric dynamics on the change in snow accumulation as well. This is further supported by other proxy evidence from central Greenland (Kapsner et al., 1995) and is similarly observed in West Antarctica (Fudge et al., 2016). An increase of the advection of air masses from lower latitudes would result in an increase of the snow accumulation as more moisture is transported to the ice sheet but in unchanged sea-salt concentrations in the precipitation and subsequently the ice core, as these air masses still carry the same amount of sea-salt aerosol. This leads to a decoupling of changes in the sea-salt aerosol deposition flux vs. the concentrations in the ice i.e., the deposition flux per event stays constant but the number of precipitation events increases. In turn, this means that the covariance of snowfall on the sea-salt deposition that is apparent on centennial and millennial time scales (Schüpbach et al., 2018; Markle et al., 2018) is less important at the interannual scale, where atmospheric dynamics dominate over thermodynamic changes. This view of changed precipitation frequency at the very beginning of the DO-onset is supported by the apparent asynchrony of snowfall and $\delta^{18}$O. Similar to Na$^+$, $\delta^{18}$O remains unchanged if only the frequency of precipitation events is increased with otherwise unaltered seasonality (and thus isotopic signature). Only later in the transition do thermodynamically-driven changes in precipitation amount and seasonality affect $\delta^{18}$O (Werner et al., 2000).

The reduction of perennial sea-ice cover during DO events that is documented in proxy records (Spolaor et al., 2016; Dokken et al., 2013) has a large influence on Greenland temperature and accumulation due to the exposure of the warm ocean in the North Atlantic as shown by model experiments (Li et al., 2010, 2005). The increased moisture availability over the open ocean which leads to the snowfall in Greenland likely also decreases the transport efficiency of marine aerosol to the Greenland ice sheet due to the increased precipitation and wash-out en route. In turn, this leads to the observed synchronous start in the transitions in ice $\delta^{18}$O and Na$^+$ concentrations. This is in good agreement with transient simulations of DO-events (Vettoretti and Peltier, 2015, 2018) showing that as soon as the North Atlantic sea-ice cover is reduced, both evaporation and precipitation over the ocean as well as temperatures in Greenland rapidly increase. The decrease of the sea-ice cover is also expected to cause a coinciding increase in snowfall. However, the increase in snow accumulation as observed in the annual layer thickness

occurs slightly earlier than the changes in sea-salt aerosol and $\delta^{18}$O, indicative of an initial increase of the advection of marine air masses to the ice sheet at the onset of the DO events unrelated to changes of the sea-ice cover. A plausible cause for this change in advection would be a shift in the jet-stream location and thus in the track of the low-pressure systems over the North Atlantic (e.g. Wunsch, 2006).

5    Mineral dust, like sea-salt aerosol, is deposited very efficiently during snowfall events because small amounts of precipitation remove most of the aerosol load from the air column (Davidson et al., 1996; Zhang et al., 2014). That means that an increase in snowfall frequency would not affect the observed dust concentration in the ice, despite the changing snow accumulation. As pointed out by Schüpbach et al. (2018), on centennial to millennial time scales, changes in the Central Asian dust source strengths by a factor of approximately 4 are needed to explain the complete amplitude between stadial and interstadials observed in the ice cores, as further supported by downwind sediment records (Porter and Zhisheng, 1995; Jacobel et al., 2017). The deflation of the dust from the Central Asian source regions is strongly dependent on location of the westerly jet (Nagashima et al., 2011; Roe, 2009) that under current conditions co-varies with the East Asian Summer Monsoon on seasonal time scales (Schiemann et al., 2009). During the glacial conditions, the westerly jet was located further south than today leading to more frequent or even permanent conditions for dust emission (Chiang et al., 2015; Nagashima et al., 2011). As low-latitude proxy records indicate northward movements of the ITCZ during the DO events (Wang et al., 2004; Deplazes et al., 2013; Wang et al., 2008; Yancheva et al., 2007), synchronous within approximately 180 yr (Adolphi et al., 2018), we hypothesize that the accompanying change in atmospheric circulation causes a reduction of the dust deflation during interstadials. Because the reduction in mineral dust aerosol concentrations in the ice core occurs before the reduction in sea-salt aerosol, the data implies that this change in atmospheric circulation happens before the reduction in the sea-ice cover in the North Atlantic. This is further supported by the fact that changes in deuterium excess at the onset of GI-1 and GI-8c, indicating changes in in moisture transport from the North Atlantic to Greenland, occur after the onset of the transition in mineral dust aerosol (Steffensen et al., 2008; Thomas et al., 2009). Antarctic records of deuterium excess that indicate changes of moisture sources also show a rapid shift at the onset of the Greenland interstadials interpreted as a change in the Southern Hemisphere westerlies explaining part of the signal observed in multiple Antarctic $\delta^{18}$O records (Markle et al., 2016; Buizert et al., 2018). This further supports the hypothesis of rapidly changing atmospheric circulation during the DO events, not only in the Northern Hemisphere, but globally. Nevertheless, the fact that for all transitions the inferred timing difference relative to the transition in $Na^+$ is smaller than the duration of the transition in that parameter indicates that the respective parts of the climate system co-evolved over the transition. That means that the changes in atmospheric circulation at the DO-onset where not completely decoupled from the change in sea-ice cover and Greenland temperature.

30    Due to the small timing differences between the records, it is worth noting, that water isotope records from polar ice cores are subject to smoothing by diffusion. For the NGRIP isotope record the diffusion length at the end of the last glacial is on the order of five to ten centimeters (Gkinis et al., 2014), influencing the high-frequency variability. For the analysis here, the diffusion means that the rapid increase of the $\delta^{18}$O signal at the DO onsets would be slightly shifted towards earlier times, leading to lower apparent lags between the aerosol records and the water isotope record. Thus, the inferred lead of $Ca^{2+}$ over
35  $\delta^{18}$O can be regarded as a conservative estimate.

## 5 Conclusions

The multi-proxy records of the timing differences from two ice cores at the onset of the DO events allow us to investigate the temporal progression of Northern Hemisphere environmental change at the onset of the events in high temporal resolution. Even though the inferred timing differences carry large uncertainties for single events, they are overall consistent between the two investigated ice-core records and over all DO events indicating their robustness. The qualitative agreement between the DO events allows for the combination of the timing differences from the individual DO events to estimate a better constrained succession of the environmental changes during the onset of the DO events. Both the initial increase in snow accumulation and the reduction of $Ca^{2+}$ occurs about a decade before the reduction in $Na^+$ and the increase in local temperature. Taken at face value, this sequence of events suggests that the collapse of the North Atlantic sea-ice cover may not be the initial trigger for the DO events and indicates that synoptic and hemispheric atmospheric circulation changes started before the reduction of the high-latitude sea-ice cover that ultimately coincided with the Greenland warming. The progression of environmental changes revealed in the Greenland aerosol records provides a good target for climate models explicitly modeling both water isotope and aerosol transport that aim at transiently simulate DO events.

Using our ice-core data, we can only derive the relative timing of changes in the atmospheric circulation and the North Atlantic surface conditions related to the DO-warming. Our results cannot provide a direct constraint on the increase in AMOC which may potentially precede changes in sea-surface conditions and atmospheric circulation. Recent studies using marine sediment records (e.g. Henry et al., 2016) suggest a centuries-long lead of AMOC changes, however the response time of the proxies, their limited resolution and dating precision may not allow an unambiguous answer yet. This question requires further studies, especially in light of model studies both with (Pedro et al., 2011, 2018) and without freshwater hosing (i.e. Zhang et al., 2014; Kleppin et al., 2015; Peltier and Vettoretti, 2014) in the North Atlantic that show that the temperature and precipitation response in Greenland is synchronous to AMOC changes.

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

## Appendix A:  Transition model

The transition between cold and warm states is described by a linear ramp of amplitude $\Delta y$, starting at $t_0, y_0$ lasting for a time of $\Delta t$:

$$\hat{y}_i(t_i) = \begin{cases} y_0 & t_i \leq t_0 \\ y_0 + \frac{t_i - t_0}{\Delta t} \Delta y & t_0 < t_i < (t_0 + \Delta t) \\ y_0 + \Delta y & t_i \geq (t_0 + \Delta t) \end{cases}$$

$y$ in this case are the log-transformed concentrations and layer thicknesses and non-transformed relative isotope ratios.

Each observation $i$ of the transitions are made with additive error $\epsilon_i$ from an irregularly sampled AR(1) noise model to account for the multi-annual variability of the proxy records as given by

$$y_i = \hat{y}_i + \epsilon_i$$

for $i > 0$ with

$$p(\epsilon_i | \epsilon_{i-1}, t_i, t_{i-1}, \tau, \sigma) = \mathrm{N}\left(\epsilon_{i-1} e^{-\frac{t_i - t_{i-1}}{\tau}}, \sigma_\epsilon^2\right)$$

where

$$\sigma_\epsilon^2 = \sigma^2 \left(1 - e^{-2\frac{t_i - t_{i-1}}{\tau}}\right),$$

and for the first observation with

$p(\epsilon_{i=0} | \tau, \sigma) = \mathrm{N}(0, \sigma^2).$

Where $p(A \mid B)$ denotes the conditional probability of $A$ given $B$. The use of an irregularly sampled AR(1) process enables the transparent handling of missing data or sampling with less than annual resolution. The two parameters of the noise process are the marginal standard deviation $\sigma$ and the auto-correlation time $\tau$ given in the same units as the data and sampling time, respectively. Both standard deviation and auto-correlation time of the noise process are determined from the data alongside the

timing parameters of the transition and are treated as nuisance parameters. For the probabilistic inference the following priors were used:

$$p(t_0) = \mathrm{N}(0.0, 50.0^2)$$

$$p(\Delta t) = \mathrm{Gamma}(2.0, 0.02)$$

$$p(y_0) = 1.0$$

$$p(\Delta y) = \mathrm{N}(0.0, 10.0^2)$$

$$p(\tau) = \mathrm{Gamma}(2.5, 0.15)$$

$$p(\sigma) \propto \frac{1}{\sigma}; \sigma < 10$$

With the following distribution functions:

$$N(\mu, \sigma^2) = p(x|\mu, \sigma^2) = \frac{1}{\sqrt{2\pi}\sigma} \exp\left(-\frac{1}{2\sigma^2}(x-\mu)^2\right)$$

$$\mathrm{Gamma}(\alpha, \beta) = p(x|\alpha, \beta) = \frac{\beta^\alpha}{\Gamma(\alpha)} x^{\alpha-1} e^{-\beta x}$$

where $\Gamma$ is the Gamma function.

Inference was performed using an ensemble Markov Chain Monte Carlo (MCMC) sampler (Goodman and Weare, 2010) implemented in the Python programming language (Foreman-Mackey et al., 2013). Samplers were run for $60\,000$ iterations using 60 ensemble members, using every $600^{\mathrm{th}}$ sample, resulting in 6000 posterior samples of the 6 parameters determining the transition and noise model. The noise model parameters were not interpreted and treated as nuisance parameters.

Because the model explicitly accounts for variability around the transition model in the data, it is possible to assess the
impact of the magnitude of this variability on the precision of the inferred timings. Intuitively, the larger the multi-annual variability in the data set compared to the amplitude of the transition, the more uncertain the inferred timing estimates should be. To test this, Figure A1 shows the marginal posterior standard deviation of the inferred onset of the transition as the Signal to Noise Ratio (SNR) as defined by the ratio of the amplitude of the transition over the standard deviation of the noise term. It shows that the higher the SNR, the lower the posterior standard deviation of $t_0$. Or in different terms: The clearer the transition
is visible, the more precise the timing inference. Because the uncertainties are propagated to the lead/lag calculations and the combined estimated we take this into account. This is becomes clear when plotting the leads/lags relative to $Na^+$ as a function of the SNR in the respective parameters as shown in Figure A2. The Figure shows no systematic connection between SNR and inferred lags.

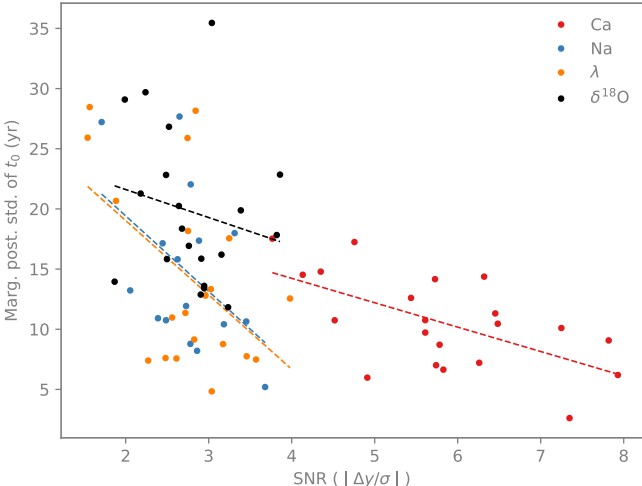

**Figure A1.** Timing uncertainty as a function of SNR for the NGRIP datasets. The uncertainty in the onset of the transition increases with lower SNR as defined by the amplitude of the transition over the amplitude of the noise. Note that, the linear fits are only for illustration.

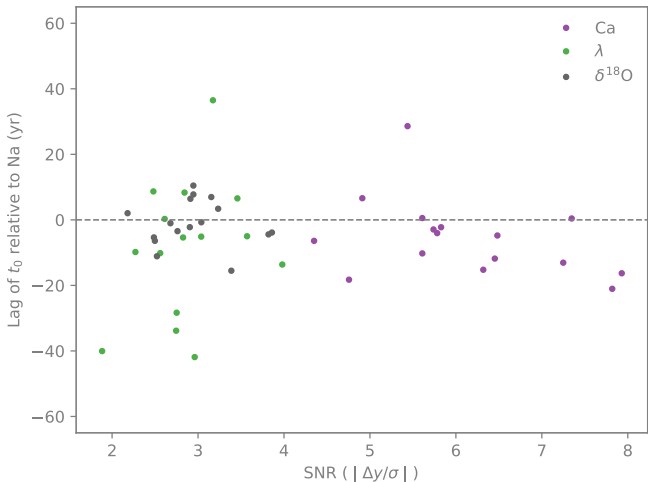

**Figure A2.** Lag relative to the transition onset in $Na^+$ vs. SNR in the respective parameters for the NGRIP data. None of the parameters show a systematic connection of the inferred leads and lags with their SNR.

*Author contributions.* M.B., S.S. and H.F. performed CFA measurements in the field at NGRIP and/or NEEM. M.B. and S.S. carried out raw data analysis. T.E. and H.F. designed the study. T.E. carried out the data analysis which was developed by T.E. in exchange with S.O.R. All authors discussed the results and contributed to the interpretation and to the manuscript, which was written by T.E.

*Competing interests.* The authors declare no competing interests.

*Acknowledgements.* The Division for Climate and Environmental Physics, Physics Institute, University of Bern acknowledges the long-term financial support of ice-core research by the Swiss National Science Foundation (SNSF) under the project numbers 200020_172506, 20FI20_137635, 10FI21_119612, 200020_159563 and 20-063333.00 as well as by the Oeschger Center for Climate Change Research. E. C. was funded by the European Union's Seventh Framework Programme for research and innovation under the Marie Skłodowska-Curie grant agreement no 600207. S. O. R. and E. C. gratefully acknowledges the Carlsberg Foudation for support to the project ChronoClimate.
F.A. is supported through a grant by the Swedish Research Council (Vetenskapsrådet no. 2016-00218). NGRIP is directed and organized by the Department of Geophysics at the Niels Bohr Institute for Astronomy, Physics and Geophysics, University of Copenhagen. It is supported by funding agencies in Denmark (SNF), Belgium (FNRS-CFB), France (IPEV and INSU/CNRS), Germany (AWI), Iceland (RannIs), Japan (MEXT), Sweden (SPRS), Switzerland (SNF) and the USA (NSF, Office of Polar Programs). NEEM is directed and organized by the Centre of Ice and Climate at the Niels Bohr Institute and US NSF, Office of Polar Programs. It is supported by funding agencies and institutions
in Belgium (FNRS-CFB and FWO), Canada (NRCan/GSC), China (CAS), Denmark (FIST), France (IPEV, CNRS/INSU, CEA and ANR), Germany (AWI), Iceland (RannIs), Japan (NIPR), South Korea (KOPRI), The Netherlands (NWO/ ALW), Sweden (VR), Switzerland (SNF), the United Kingdom (NERC) and the USA (US NSF, Office of Polar Programs) and the EU Seventh Framework programmes Past4Future

and WaterundertheIce. The authors also gratefully acknowledge the contributions of the countless people that facilitated and took part in both the ice-core drilling and processing as well as the CFA melting campaigns. The authors would further like to thank the reviewers for their use full comments that helped to improve the manuscript.