# Peer review of "Decadal-scale progression of the onset of Dansgaard-Oeschger warming events"

_Climate of the Past, 2018_

## Referee Comment (RC1) · Anonymous Referee #1 · 5 Feb 2019

Review of Erhardt et al: Decadal-scale progression of Dansgaard-Oeschger warming events

Erhardt et al. present high-resolution Ca and Na records from the Greenland NGRIP and NEEM ice cores, and combine these with NGRIP annual layer thickness (an estimate of past accumulation) and d18O data. The study evaluates the phasing of the various records during abrupt climate events of the Dansgaard-Oeschger cycle, and finds a clear sequence of events consistent with some previous work. Accumulation rate changes lead, followed by Ca, d18O and finally Na. Taken at face value, this sequence would argue for event initialization at lower latitudes, with the sea ice response associated with DO events coming last.

These records are of great value to the scientific community, and the analysis is meaningful and appears to be rigorously done (request for minor clarification below). This paper is clearly a great contribution to the literature, and I have only some minor suggestions that may improve the clarity and interpretation.

(1) The current manuscript only describes the relative phasing of the onset, midpoint and endpoint of each transition. What is missing is an analysis of the duration of each of the transitions, to put the lead/lag values into perspective. For example, if the transition were to take 100 years, then a 10-yr lead indicates that the climatic components reflected by these records co-evolved; if the transition were to take 5 years only then the same 10-yr lead suggests a decoupling, with the shift in one component (say the jet) completed before the others respond.

I request the authors add one panel to Figures 3 and 4 each that gives the transition duration.

Many of us are visually oriented. Would it be possible to show the D-O average transition in Ca, Na, lambda and d18O together in one plot like is done in Fig. 2 for Ca? Either the data or just the fits (if the data are too messy). This would really give a nice visual representation of how the "average" transition occurs.

(2) The paper only analyzes the glacial-to-interglacial (or D-O warming) transitions, and not the interglacial-to-glacial (or D-O cooling) transitions. Have you tried analyzing the latter? I would be very interested to see the phasing for these transitions also. I imagine it is more challenging given the smaller and less abrupt nature of these transitions – but I think it would be valuable nevertheless.

(3) How precise is the depth registration of the various CFA components relative to one another? And relative to the layer thickness and d18O records? I can imagine there may be cm-scale offsets, which could become important given the extremely small time phasings that the authors interpret. Please address this briefly.

(4) Appendix A was not completely clear to me, and I think it should be elaborated on
in some more detail for the lay reader. Do I understand correctly that the function that is being fit to the data is the ramp function plus an AR(1) noise function, and that the parameters of both are varied in the Monte Carlo sampler?

The language of prior and posterior distributions suggests this is a Bayesian approach – please confirm.

How is the goodness of fit evaluated, and what criterion is used in the MC sampler to accept or reject individual solutions?

It would help greatly if you could add a figure showing how several individual iterations of the fitting process look like.

(5) I am confused by the statements below Fig 4 (Starting on P9 L18). If both Ca and lambda lead Na by  $\sim$ 10 years, how come these two are not necessarily synchronous? This is very counter-intuitive; all records are evaluated on the same depth scale, so why wouldn't they be? I think the relative phasing of all these records is the central result of this paper, so it would be important to establish a robust sequence of events. What would be needed to establish synchroneity of Ca and lambda? Would you need to run the analysis again relative to the Ca transition instead of relative to the Na transition? If not too much work, that may be worth doing, given the importance for the interpretation.

I could imagine a 4x4 matrix for NGRIP with the lead/lag of each of the records evaluated relative to the others, and a 2x2 matrix of the same for NEEM.

(6) All age axes have a "BP" label. Do you use BP 1950 or b2k? This is a contentious point in the ice core community (Wolff, 2007), but BP 1950 is the best choice in my view based on precedent in the literature and convention of the radiometric dating communities. At least specify which is used.

Line-by-line comments:

P1 L17-19: The phrase "DO event" is unfortunately ambiguous, with some people equating them to the abrupt warming phases, and others to the interstadials. To
avoid this, consider "... millennial-scale abrupt climate change, called the Dansgaard-Oeschger cycle (REFS). During abrupt DO warming, ..."

L22: I assume your are still talking about DO warming here? Please specify that the changes described are for the warming phase.

P2 L6: Consider replacing Henry et al. with the recent review by (Lynch-Stieglitz, 2017), to avoid arbitrarily picking one study out of dozens that demonstrate the link to AMOC.

P2 L23: "Some of ...". I think you can safely say "All of" (or "most of " to be conservative). I am not aware of any model simulation or theory that does not involve sea ice as either the trigger or amplifier. You simply cannot get that much warming that quickly without sea ice.

P3 L20: "exact co-registration". How exact? Please specify relative and absolute depth registration of various CFA components.

P3 L23: Do you use the actual single layer annual counts, or the 20yr averages that are publicly available?

P3 L27: All data ARE shown... (the word "data" is plural not singular)

Figure 1+3: Please consider plotting the age axis in reverse (so time goes from left to right). This is what much of the paleoclimate literature is moving towards. That is also what Figure 2 uses.

P5 L30-31: So are your interpreting the changes in terms of the source strength only? Or does transport to the site dominate? Some would argue for the latter.

P5 L33: Do you actually fit an exponential, or do you fit a linear to the log(Ca) time series?

P6L16: "decreases" should be "increases" here, right? (it goes from 2 to 7yr, so increase?)

CPD
Figure 2: please add units to the y-axis labels. Also, how come the fit is so smooth / rounded? Isn't the fit a linear ramp? Is this because you average over many solutions?

P9 L5: Buizert et al. 2015 should be cited as WAIS Divide Project Members 2015.

P10 L23-24: Add a reference for this claim.

P10 L24: "lack of covariance" seems like strange phrase here. The records you are talking about are correlated with r > 0.95 probably.

P10 L32: "events" is confusing here. Are you talking about individual synoptic / precip events? or DO events, better specify more clearly.

P11 L15: This idea was suggested by (Seager and Battisti, 2007) and (Wunsch, 2006)

P11 L16: I had expected a larger discussion about wet vs. dry deposition. Could the coincidence of lambda and Ca changes be explained that way to some degree?

P11 L18: effect should be affect

P11 L34: This further supportS...

P12 L13: "reduction of the sea ice cover that ultimately coincided with the Greenland warming AND WAS PRESUMABLE A MAJOR DRIVER THEREOF"

Again, I think it's hard (impossible?) to get such a large Greenland temp response without a change in sea ice cover.

P18 L25: What is the rationale for taking the log of lambda instead of just lambda itself?

P19: specify what all the symbols mean in your maths.

References:

Lynch-Stieglitz, J., 2017. The Atlantic Meridional Overturning Circulation and Abrupt Climate Change. Annual Review of Marine Science 9, 83-104.

Seager, R., Battisti, D.S., 2007. Challenges to our understanding of the general circu-
lation: abrupt climate change. Global Circulation of the Atmosphere, 331-371.Wolff, E.W., 2007. When is the "present"? Quat. Sci. Rev. 26, 3023-3024.Wunsch, C., 2006. Abrupt climate change: An alternative view. Quat. Res. 65, 191-203.

---

## Referee Comment (RC2) · Bradley Markle (Referee) · 8 Mar 2019

In this study the authors use high-resolution ice-core records of aerosols, water isotopes, and layer thickness from Greenland to examine phasing of different aspects of the climate system during Dansgaard-Oeschger events. They use objective techniques to tease out small leads and lags between the noisy records, showing that changes in calcium aerosols and layer thickness lead changes in sodium aerosols and water isotope ratios. They conclude that these lags suggest that changes in sea ice extent did not occur before other changes in the climate system, namely atmospheric circulation. The manuscript is very well written, the analysis is careful, and the discussion is well-argued. The results are quite impressive and should be of wide interest to the community. I have a couple questions and concerns that I hope the authors will address

and then several minor questions.

(somewhat) Major questions: The authors have made a compelling observation through their analysis of leads and lags between the proxy records. They attribute these differences in timing to aspects of the climate system that have different influence on the proxies. My first two questions have to do with whether these leads and lags could arise from other factors. I suspect that these concerns are not too important to the conclusions of this study.

1. How does the signal-to-noise ratio of the record influence fitting the transition model and the identification of starting, mid, and end points? Here I'm thinking of the SNR as quantified by the size of the transitions compared to the variance within the stadials and interstadials. I realize the fitting procedure takes into account the interannual noise and its autocorrelation. But if you have an idealized known ramp function with different levels of background noise, will the model find the same starting, mid, and endpoints?

One could imagine that an increased background noise could lead to the identification of time-shifted transition points depending on the fitting technique. One then risks conflating difference in the timing of signals between records with difference in one's ability to detect signals between records. I cannot tell from the description of the transition model alone how much of an issue this is to this analysis. Doing some simple tests with a couple different ramp-fitting and significant change detection techniques (though ones less sophisticated than the technique used by the authors), I find that different levels of interannual noise can influence the timing of a fitted transition, though not in all circumstances.

My particular worry here is that the substantially higher noise (interannual variability) in the Na records could lead to the identification of a delayed onset or shorter transitions compared to the Ca and other records. My worry is heightened slightly in that the relative timing seems to correspond to the level of background noise (at least visually): the d18O and Na timing are most similar among the proxies and also both appear to

have much lower SNR (more noise) compared to the Ca and layer thickness records.

Do the mean lags depend on the amplitude of the DO event or the length of the transition between stadial and interstadial? This could be the case if the ramp-fitting depends on the amplitude of back ground noise. I realize this could be hard to determine since one needs to look at many events at once to see the mean lags. But a scatter plot of lags vs. event magnitude or fitted ramp duration could be informative.

I suspect that this concern is entirely accounted for by the very careful analysis of the marginal posterior densities of the onsets, midpoints, and endpoints for each proxy and the comparison between proxies, that the authors have already performed. It would however be helpful to see the influence, or the demonstration of the absence of influence, of the SNR on the fitting procedure given that the conclusions rest on the difference in timing with respect to Na in particular. I'd find it very useful to see this demonstrated on artificial ramp signals (of varying duration) where the true timings are known explicitly, with varying SNR, and especially with the SNRs relevant to d18O, lambda, Ca, and Na. It seems crucial to know that different SNR alone can not account for the difference in timing identified by the fitting procedure.

2. Can the authors rule out the influence of water isotope diffusion on the difference in timing of the d18O signals and those of Ca and layer thickness? Based on analysis of NGRIP (Gkinis et al 2014) and a similar site in Antarctica (Jones et al 2017, 2018), I'd guess diffusion lengths are on the order of 5-10cm through this interval and so are not insignificant compared to the annual layer thickness. Such diffusion lengths can have meaningful influence on the inter-annual and even decadal variability in the water isotope record (Jones et al 2018). I imagine that correcting the records used here for the potential influence of diffusion, if that would even be sensible, is far beyond the scope of this study. However, it seems entirely reasonable to estimate the influence (if any) of the smoothing implied by diffusion on the timing identified by the transition model fitting procedure. If you take identical idealized ramps, and smooth one with a time-scale reflective of water isotope diffusion lengths, will the fitting procedure identify

the same start, mid, and end points? I suspect these effects, if any, are small, though we are only talking about lags of a few years.

3. The most interesting conclusion of this study to my mind is the authors statement that "at face value, this sequence of events suggests that the collapse of North Atlantic sea-ice cover is not the initial trigger for the DO events..." Because of its potential wide interest, this statement deserves some scrutiny. It rests on the authors use of Na as a "qualitative indicator of sea ice cover" in the North Atlantic.

The discussion on Page 5, lines 3-28, highlights the debate over in the interpretation of Na very well. The authors interpretation is laid out on Page 5 lines 20-25. Markle et al 2018 find that most of the millennial variability in Antarctic sea salts can be explained simply by the changes in moisture rainout that are required to explain the water isotope record (these changes also explain most of the changes in Antarctic Ca variability and its relationship to water isotopes in both Antarctica and Greenland (c.f. their Figure 4)). This suggests comparatively small if any changes in sea salt source latitude or strength are needed to explain those Na records (though changes in those things are still possible of course). Is there evidence that this explanation for the sea salts is insufficient in Greenland? Are the observed changes in sea salt for example much larger than what one would expect from temperature dependent rainout alone? It was unclear to me from this discussion that the sea salt source strength (or even mean source latitude) should have a clear relationship to the sea ice edge.

If the main way sea ice influences Greenland Na is through its relationship to the variables driving the rainout effect, then a change in sea ice extent doesn't necessarily mean one should have a coincident change in Greenland Na, particularly at interannual timescales.

For example, one can imagine a scenario in which sea ice extent begins to retreat at exactly the same time as the changes observed in Ca initiate. Coincident increases in temperature and moisture removal would cause a decrease in the amount of Na

(and Ca) reaching Greenland (as described by the authors). However, if that change in sea ice extent caused an increase in sea salt source production or a northward migration of mean source latitude (both debatable but reasonable, particularly the latter) this could temporally compensate for the increased removal. This combination of influences could lead to an apparent timing difference in the final Na signal observed in Greenland compared to the actual timing of sea ice changes (an example of the superposition of competing source and rainout factors on polar aerosols is given in the Supplement of Markle et al 2018 c.f. Figures S9). A somewhat similar scenario may be likely for the water isotopes, as changing sea ice extent could drive moisture source effects that could temporarily compensate for the decreased depletion driven by simultaneous changes over the ice sheet (these would be of the correct sign to compensate, though it would be hard to assess the potential size of this effect on the isotopes without analyzing deuterium excess records from the same cores). Quantitatively disentangling these influences may be well outside the scope of this study. But this does at least suggest a limitation to using Na as a qualitative indicator of sea ice, and should suggest some commensurate caution in the conclusions drawn based on that interpretation.

Even in the absence of competing influences, uncertainty in the linearity of the sodium-as-sea-ice-extent interpretation poses challenges. Dose a change in the sea ice edge (or extent) of a given size have the same impact on Na in Greenland if the sea ice edge is at 55 degrees North (just for example) versus if the edge is at 65 degrees North? If relationship between changes Greenland Na and the sea ice absolute position is (sufficiently) nonlinear then the changes in sea ice at the start of DO events may not be as detectable in Greenland Na as changes later in the event. This could lead to apparent lags of the Na signal to Ca, even if the change in sea ice initiates at the same time as the change in whatever drives Ca. I'm not sure I think this is particularly likely, but it is certainly plausible. Further, the impact of sea ice on climate isn't linear. Again a given size change in the sea ice edge when the edge is at 55 degrees North (for example) doesn't have the same impact on the surface radiation budget nor the

buoyancy forcing of the overturning circulation as when the edge is at 65 degrees North. The relative timing of sodium changes in Greenland don't necessarily rule out sea ice changes as a potential "trigger" of DO events, if climatically meaningful sea ice changes can happen without influencing Greenland Na.

To be sure, there are innumerable, potentially ad hoc, explanations for the data, and it is not the authors responsibility to come up with and then weigh the merits of all such explanations. However the assertion that the lag in sodium seen in Greenland necessitates a lag in changes in sea ice extent with respect to shifts in atmospheric circulation, and that this in turn rules out sea ice changes as the trigger for DO events, is somewhat bold. That this "provides an essential benchmark for climate models" is bolder still. Both of these statements to my mind somewhat outpace the robustness in the interpretation of Na as a qualitative indicator of sea ice extent. I think some slight tempering of the conclusions is merited here and perhaps some discussion of the limitations to the interpretation, or much more compelling evidence is needed that a change in sea ice extent must be seen in the Greenland Na records regardless of other processes. To be clear, I do think the difference in timing of the aerosol species identified here is a compelling target for modeling. And I do think the climatic interpretation offered by the authors is plausible, and one to be taken seriously. It is just not clear to me that these data alone place that strong of constraints on the timing of sea ice changes.

Minor points: Page 1, line 8: The authors say "from one of the cores".... Which one?

Page 1 line 17: I think that the clause "In the course of the last glacial period" should be moved after the 2nd comma in this line (after "warming episodes,"). This is a possibly silly language thing but: the ice core records reveal that the events were in the last glacial period. It wasn't during the last glacial period that the ice core records revealed these events (that was in the 1980s!).

Page 1 line 20: This is a language choice the authors should feel free to ignore but

"..going along with an almost doubling..." sounds strange to my ear, though I can't tell why. "...coincident with a near-doubling..." sounds better to me.

Page 2 line 3: I'd move the "Also" at the start of the sentence to after "Northern Hemisphere".

Page 2 Lines 20-35: This is a very nice description of several related though distinct ideas. Thanks!

Page 3 lines 15-20: Are the Ca and Na records corrected for the sea-salt vs non sea-salt components of each (e.g. using average Na/Ca mass ratios in average crust and sea salt, as is common in Antarctic records)? I ask because having looked into this once briefly, it seemed like the corrections often used in Antarctica lead to nonsensical results in Greenland, which was disturbing, and I wasn't sure why.

Page 6, Line 9: I think there is a missing "of" between "interpretation" and "phase".

Page 6, lines 20-21: This is interesting! Nice.

Page 6, lines 27: "on" should be "one" I think.

Page 6, lines 29: If you don't make the assumption that the DO-events have the same imprint in both cores, how would that effect any of your conclusions?

Page 10, lines 25-30: Might be worth citing Fudge et al 2016 here, they show a strong divergence of accumulation rate and d18O in ice cores at millennial time scales.

Page 11, lines 3-5: This is nice, convincing analysis.

Page 11, lines 11: I think this should be "...a coinciding..." Not "...an coinciding..."

Page 11, lines 21-22: How big are the additional changes in source need compared to the total variability across a DO event?

Page 11, lines 24-35: This is really nice discussion.

Refs cited: Markle, Bradley R., et al. "Concomitant variability in high-latitude aerosols,

water isotopes and the hydrologic cycle." Nature Geoscience (2018): 1.

Jones, Tyler R., et al. "Southern Hemisphere climate variability forced by Northern Hemisphere ice-sheet topography." Nature 554.7692 (2018): 351.

Jones, T. R., et al. "Water isotope diffusion in the WAIS Divide ice core during the Holocene and last glacial." Journal of Geophysical Research: Earth Surface 122.1 (2017): 290-309.

Fudge, T. J., et al. "Variable relationship between accumulation and temperature in West Antarctica for the past 31,000 years." Geophysical Research Letters 43.8 (2016): 3795-3803.

Gkinis, Vasileios, et al. "Water isotope diffusion rates from the NorthGRIP ice core for the last 16,000 years–Glaciological and paleoclimatic implications." Earth and Planetary Science Letters 405 (2014): 132-141.
* * *

---

## Author Comment (AC1) · 5 Apr 2019

**Reply to reviewer comment #1**

Review of Erhardt et al: Decadal-scale progression of Dansgaard-Oeschger warming events

Erhardt et al. present high-resolution Ca and Na records from the Greenland NGRIP and NEEM ice cores, and combine these with NGRIP annual layer thickness (an estimate of past accumulation) and d18O data. The study evaluates the phasing of the various records during abrupt climate events of the Dansgaard-Oeschger cycle, and finds a clear sequence of events consistent with some previous work. Accumulation rate changes lead, followed by Ca, d18O and finally Na. Taken at face value, this sequence would argue for event initialization at lower latitudes, with the sea ice response associated with DO events coming last.

These records are of great value to the scientific community, and the analysis is meaningful and appears to be rigorously done (request for minor clarification below). This paper is clearly a great contribution to the literature, and I have only some minor suggestions that may improve the clarity and interpretation.

> Thank you!

(1) The current manuscript only describes the relative phasing of the onset, midpoint and endpoint of each transition. What is missing is an analysis of the duration of each of the transitions, to put the lead/lag values into perspective. For example, if the transition were to take 100 years, then a 10-yr lead indicates that the climatic components reflected by these records co-evolved; if the transition were to take 5 years only then the same 10-yr lead suggests a decoupling, with the shift in one component (say the jet) completed before the others respond.

I request the authors add one panel to Figures 3 and 4 each that gives the transition duration.

> We agree that this is an important point. Because both Figure 3 and 4 show difference relative to Na, we prefer not to add absolute information to these figures. However, to answer the reviewer's comment, we have added the relevant information to the text (P 7, L 11 ff):
> "For all of the transitions, the inferred timing differences relative to the onset of the transition in Na are smaller than the duration of the transition itself in each of the proxy records.
> That means that none of the proxies exhibit a complete stadial-interstadial transition before the onset of the transition in the sea-salt aerosol concentration."
> And later in the discussion (P 12, L 25 ff):
> "The fact that for all transitions the inferred timing difference relative to the transition in Na is smaller than the duration of the transition in that parameter indicates that the respective parts of the climate system co-evolved over the transition. That means that the changes in atmospheric circulation at the DO-onset where not completely decoupled from the change in sea-ice cover and Greenland temperature."

Many of us are visually oriented. Would it be possible to show the D-O average transition in Ca, Na, lambda and d18O together in one plot like is done in Fig. 2 for Ca? Either the data or

just the fits (if the data are too messy). This would really give a nice visual representation of how the "average" transition occurs.

Because all of the transitions have different lengths it is not possible to meaningfully stack them. This is the reason why we chose to combine the estimates of the relative onset timings and not the complete transitions. We would like to point out that Figure 2 only shows the fit of a single transition, not a stack.

(2) The paper only analyzes the glacial-to-interglacial (or D-O warming) transitions, and not the interglacial-to-glacial (or D-O cooling) transitions. Have you tried analyzing the latter? I would be very interested to see the phasing for these transitions also. I imagine it is more challenging given the smaller and less abrupt nature of these transitions but I think it would be valuable nevertheless.

We agree with the reviewer that analyzing the cooling transitions would be very valuable. However, in the presented manuscript, we deliberately chose to focus on the DO onsets as they are much more clearly distinguishable in the records and are possible to describe using the relatively simple model employed here. Furthermore, the fundamental difference between the warming and cooling transitions in their manifestation in the records suggests a different mechanism and thus merits its own separate investigation which we plan to do in the future.

(3) How precise is the depth registration of the various CFA components relative to one another? And relative to the layer thickness and d18O records? I can imagine there may be cm-scale offsets, which could become important given the extremely small time phasings that the authors interpret. Please address this briefly.

Between the different CFA records, the depth assignment is accurate to within a few mm as they are measured on the same piece of ice and the relative component alignment is measured regularly as described in the cited references. Because the NGRIP CFA records form the data basis for the annual layer-counting, the CFA depth scale is the fundamental depth scale to which the layer thicknesses and dating refers. The relative uncertainty of the depth assignment of the CFA/layer thickness data to the d18O record is fundamentally limited by the accuracy of the subsampling within each 55 cm piece of ice (called a bag). Typically, sample depths are measured with a meter stick and are accurate to within half a centimeter or better.
We added the relevant information to the manuscript. (P 3 L 29 ff):
"Between the individual data sets the co-registration uncertainty is limited by the absolute depth assignment of the data sets. This uncertainty is typically on the order of a few millimeter to at most single centimeters and thus on the which translates to a co-registration uncertainty in the sub-annual range."

(4) Appendix A was not completely clear to me, and I think it should be elaborated on in some more detail for the lay reader. Do I understand correctly that the function that is being fit to the data is the ramp function plus an AR(1) noise function, and that the parameters of both are varied in the Monte Carlo sampler?

The language of prior and posterior distributions suggests this is a Bayesian approach – please confirm.

How is the goodness of fit evaluated, and what criterion is used in the MC sampler to accept or reject individual solutions?

It would help greatly if you could add a figure showing how several individual iterations of the fitting process look like.

> We believe that a thorough introduction to Bayesian statistics and Markov Chain Monte Carlo methods is beyond the scope of this paper.
> Nevertheless, we want to give an intuition on the sampling process here:
> In the sampling process, the sampler randomly walks the whole parameter space, in this case represented by the four parameters describing each transition and the two noise parameters. At each step, the sampler evaluates the likelihood of observing the data given the current set of parameters and accepts or rejects these parameters with a probability proportional to this likelihood. If performed over a large number of samples, this process will yield samples from the posterior distribution which is what we use for our analysis (see for example Gelman et al., 2013)
>
> Gelman, A. et al. (2013), Bayesian data analysis, Chapman and Hall/CRC

(5) I am confused by the statements below Fig 4 (Starting on P9 L18). If both Ca and lambda lead Na by ~10 years, how come these two are not necessarily synchronous? This is very counter-intuitive; all records are evaluated on the same depth scale, so why wouldn't they be? I think the relative phasing of all these records is the central result of this paper, so it would be important to establish a robust sequence of events. What would be needed to establish synchroneity of Ca and lambda? Would you need to run the analysis again relative to the Ca transition instead of relative to the Na transition? If not too much work, that may be worth doing, given the importance for the interpretation.

I could imagine a 4x4 matrix for NGRIP with the lead/lag of each of the records evaluated relative to the others, and a 2x2 matrix of the same for NEEM.

> We agree that this is indeed counter-intuitive. We slightly rephrased the sentence to make this point clearer (see below) The reason for this is that the uncertainties both for the lags in the individual transitions as well as for the combined evidence is very strongly dependent on the estimates from the layer thickness timeseries. Depending on how these highly correlated uncertainties are projected onto the different axis (lt-Na, Ca-Na, Ca-lt) slightly different distributions arise. For Ca and lt, the timing difference at the onset of the transition shows a lead of Lambda over Ca with 4 (-5/+4) yr that is not incompatible with a zero lag at the 95% probability level.
> A different way of looking at the order of events is to calculate the average probability of the order of the transition onsets. This shows that the transition Lambda and Ca are about equally likely to occur first, whereas Na and d18O are about equally likely to occur last. From this type of analysis, one can also establish the most likely order of onsets during the warming transitions which is lt, Ca, d18O, Na for NGRIP and Ca, Na for NEEM.
> We have added these results in a new Figure focusing on the NGRIP ice core into the revised manuscript (P9, L16 ff and Figure 5).
>
> "Note that the density functions shown in Figure 4 cannot be used to infer timing differences between the other parameters. This is a direct result of the estimates being

conditional on the timing of the transition in sodium, leading to large correlations between the lag estimates for the other parameters. That means that even though e.g two probability density functions of the differences relative to the transition in sodium largely overlap, that does not necessarily mean that the timing difference is equal to zero. In the case of the timing difference between the transition onsets of the increase in annual layer thickness and the decrease in $Ca^{2+}$ concentrations the combined lead of the change in annual layer thickness is not larger than zero at the 0.95 probability level with 4(-5, +4) years. To establish the most probable sequence of events at the transition offset we calculate the average order of the onset times, shown for NGRIP in Figure 5. The average positions show that the change in accumulation and $Ca^{2+}$ concentrations about equally likely occur first whereas the transitions in $Na^+$ and $\delta^{18}O$ about equally likely occur last. The same analysis for the NEEM results confirms this sequence."

(6) All age axes have a "BP" label. Do you use BP 1950 or b2k? This is a contentious point in the ice core community (Wolff, 2007), but BP 1950 is the best choice in my view based on precedent in the literature and convention of the radiometric dating communities. At least specify which is used.

We had only specified this in the caption of Figure 1, but we have now also added this to the text (P4, L1)
"All ages are given relative to 1950."
We have also changed all axis-labels to "before 1950" instead of "BP" to avoid confusion with radiocarbon ages.

Line-by-line comments:

P1 L17-19: The phrase "DO event" is unfortunately ambiguous, with some people equating them to the abrupt warming phases, and others to the interstadials. To avoid this, consider ". . . millennial-scale abrupt climate change, called the DansgaardOeschger cycle (REFS). During abrupt DO warming, . . ."

Thank you for the suggestion. To make the text more consistent we changed the wording throughout the text so to specifically state that we are looking at the onset of the D-O events. However, we deliberately do not use the term "DO cycles" to avoid any notion of cyclicity in these events.

L22: I assume your are still talking about DO warming here? Please specify that the changes described are for the warming phase.

Clarified in the text.

P2 L6: Consider replacing Henry et al. with the recent review by (Lynch-Stieglitz, 2017), to avoid arbitrarily picking one study out of dozens that demonstrate the link to AMOC.

Done.

P2 L23: "Some of . . .". I think you can safely say "All of" (or "most of " to be conservative). I am not aware of any model simulation or theory that does not involve sea ice as either the trigger or amplifier. You simply cannot get that much warming that quickly without sea ice.

We have now changed the sentence and use "Many of…". Most of the experiments are run without an interactive sea-ice model and thus cannot realistically inform on sea-ice feedbacks. We do however agree that most of the recent model studies with coupled models involve the sea ice as triggers or amplifiers.

P3 L20: "exact co-registration". How exact? Please specify relative and absolute depth registration of various CFA components.

We added the relevant information to both the CFA data sets as well as to the end of the paragraph:

"…exact co-registration of the aerosol concentration records at the millimeter scale…" (P3, L21)

"Between the individual data sets the co-registration uncertainty is limited by the absolute depth assignment of the data sets.
This uncertainty is typically on the order of a few millimeter for CFA data up to around a centimeter between d18O and CFA data, which translates to a co-registration uncertainty of sub-annual to annual range." (P3, L29 ff)

P3 L23: Do you use the actual single layer annual counts, or the 20yr averages that are publicly available?

The data presented here is based on the single annual layer counts and new 10yr averages are published alongside this paper for the whole 60 ka as well as higher resolution datasets for the individual transitions.

P3 L27: All data ARE shown. . . (the word "data" is plural not singular)

Updated accordingly.

Figure 1+3: Please consider plotting the age axis in reverse (so time goes from left to right). This is what much of the paleoclimate literature is moving towards. That is also what Figure 2 uses.

To make all axis directions consistent, we changed the axis in Figure 2 to match those in the other figures.

P5 L30-31: So are your interpreting the changes in terms of the source strength only? Or does transport to the site dominate? Some would argue for the latter.

The exact interpretation in terms of source and transport changes is described in detail in the discussion section of the paper. (P10 ff)

P5 L33: Do you actually fit an exponential, or do you fit a linear to the log(Ca) time series?

We fit a linear function to the log(Ca) and log(Na) time series, as described in the appendix. We added this also to the main text for clarification: (P6, L2 ff)

"…or exponential transition (i.e. a linear transition fitted to the log-transformed data for all other records) between two constant levels…"

P6L16: "decreases" should be "increases" here, right? (it goes from 2 to 7yr, so increase?)

We prefer decreases in this case as "high resolution" typically referrers to lower sampling intervals and vice versa.

Figure 2: please add units to the y-axis labels. Also, how come the fit is so smooth / rounded? Isn't the fit a linear ramp? Is this because you average over many solutions?

Yes, the smoothness is a feature of the marginal posterior median ramp which is shown here. The individual realizations are linear ramps with sharp kinks. We have now added units to the y-axis labels

P9 L5: Buizert et al. 2015 should be cited as WAIS Divide Project Members 2015.

We have now updated the reference accordingly.

P10 L23-24: Add a reference for this claim.

We have now added Pausata et al .2009 and Merz et al. 2013 as references.

Pausata, F. S. R. et al. (2009), Changes in atmospheric variability in a glacial climate and the impacts on proxy data: a model intercomparison, Climate of the Past, 5489-502, doi:10.5194/cp-5-489-2009.

Merz, N. et al. (2013), Greenland accumulation and its connection to the large-scale atmospheric circulation in ERA-Interim and paleoclimate simulations, Climate of the Past, 92433-2450, doi:10.5194/cp-9-2433-2013.

P10 L24: "lack of covariance" seems like strange phrase here. The records you are talking about are correlated with r > 0.95 probably.

We have now rephrased to "…lack of synchronous changes…"

P10 L32: "events" is confusing here. Are you talking about individual synoptic / precip events? or DO events, better specify more clearly.

We have now specified as "precipitation events".

P11 L15: This idea was suggested by (Seager and Battisti, 2007) and (Wunsch, 2006)

We have now added Wunsch, 2006 as original reference.

P11 L16: I had expected a larger discussion about wet vs. dry deposition. Could the coincidence of lambda and Ca changes be explained that way to some degree?

> Ca is, as sea salt, very efficiently scavenged by snowfall events and its deposition is hence governed by the frequency of the precipitation events.

P11 L18: effect should be affect

> Corrected, thanks for pointing this out.

P11 L34: This further supportS. . .

> Corrected, thanks for pointing this out.

P12 L13: "reduction of the sea ice cover that ultimately coincided with the Greenland warming AND WAS PRESUMABLE A MAJOR DRIVER THEREOF" Again, I think it's hard (impossible?) to get such a large Greenland temp response without a change in sea ice cover.

> We completely agree and we do not argue against the fact that the sea-ice change is likely the major driver of the warming in Greenland.

P18 L25: What is the rationale for taking the log of lambda instead of just lambda itself?

> We took the log of the annual layer thickness, because it is very well described by a log-normal distribution as shown in Andersen et al. 2006.
>
> *Andersen, K. K. et al. (2006), The Greenland Ice Core Chronology 2005, 15-42 ka. Part 1: constructing the time scale, Quaternary Science Reviews, 253246-3257, doi:10.1016/j.quascirev.2006.08.002*

P19: specify what all the symbols mean in your maths.

> We have now added the missing symbols to the text.

References:

Lynch-Stieglitz, J., 2017. The Atlantic Meridional Overturning Circulation and Abrupt Climate Change. Annual Review of Marine Science 9, 83-104.

Seager, R., Battisti, D.S., 2007. Challenges to our understanding of the general circulation: abrupt climate change. Global Circulation of the Atmosphere, 331-371.

Wolff, E.W., 2007. When is the "present"? Quat. Sci. Rev. 26, 3023-3024.

Wunsch, C., 2006. Abrupt climate change: An alternative view. Quat. Res. 65, 191203.

---

## Author Comment (AC2) · 5 Apr 2019

**Reply to reviewer comment #2**

In this study the authors use high-resolution ice-core records of aerosols, water isotopes, and layer thickness from Greenland to examine phasing of different aspects of the climate system during Dansgaard-Oeschger events. They use objective techniques to tease out small leads and lags between the noisy records, showing that changes in calcium aerosols and layer thickness lead changes in sodium aerosols and water isotope ratios. They conclude that these lags suggest that changes in sea ice extent did not occur before other changes in the climate system, namely atmospheric circulation. The manuscript is very well written, the analysis is careful, and the discussion is well-argued. The results are quite impressive and should be of wide interest to the community. I have a couple questions and concerns that I hope the authors will address and then several minor questions.

> Thank you!

(somewhat) Major questions: The authors have made a compelling observation through their analysis of leads and lags between the proxy records. They attribute these differences in timing to aspects of the climate system that have different influence on the proxies. My first two questions have to do with whether these leads and lags could arise from other factors. I suspect that these concerns are not too important to the conclusions of this study.

1. How does the signal-to-noise ratio of the record influence fitting the transition model and the identification of starting, mid, and end points? Here I'm thinking of the SNR as quantified by the size of the transitions compared to the variance within the stadials and interstadials. I realize the fitting procedure takes into account the interannual noise and its autocorrelation. But if you have an idealized known ramp function with different levels of background noise, will the model find the same starting, mid, and endpoints?

One could imagine that an increased background noise could lead to the identification of time-shifted transition points depending on the fitting technique. One then risks conflating difference in the timing of signals between records with difference in one's ability to detect signals between records. I cannot tell from the description of the transition model alone how much of an issue this is to this analysis. Doing some simple tests with a couple different ramp-fitting and significant change detection techniques (though ones less sophisticated than the technique used by the authors), I find that different levels of interannual noise can influence the timing of a fitted transition, though not in all circumstances.

My particular worry here is that the substantially higher noise (interannual variability) in the Na records could lead to the identification of a delayed onset or shorter transitions compared to the Ca and other records. My worry is heightened slightly in that the relative timing seems to correspond to the level of background noise (at least visually): the d18O and Na timing are most similar among the proxies and also both appear to have much lower SNR (more noise) compared to the Ca and layer thickness records.

Do the mean lags depend on the amplitude of the DO event or the length of the transition between stadial and interstadial? This could be the case if the ramp-fitting depends on the amplitude of back ground noise. I realize this could be hard to determine since one needs to look at many events at once to see the mean lags. But a scatter plot of lags vs. event magnitude or fitted ramp duration could be informative.

I suspect that this concern is entirely accounted for by the very careful analysis of the marginal posterior densities of the onsets, midpoints, and endpoints for each proxy and the comparison between proxies, that the authors have already performed. It would however be helpful to see the influence, or the demonstration of the absence of influence, of the SNR on the fitting procedure given that the conclusions rest on the difference in timing with respect to Na in particular. I'd find it very useful to see this demonstrated on artificial ramp signals (of varying duration) where the true timings are known explicitly, with varying SNR, and especially with the SNRs relevant to d18O, lambda, Ca, and Na. It seems crucial to know that different SNR alone can not account for the difference in timing identified by the fitting procedure.

We do completely agree with this concern and thank you for bringing this up. You are correct in that the signal to noise ratio as defined by the amplitude of the ramp divided by residual standard deviation varies widely between the different parameters and that Ca has by far the highest SNR of the records. In our analysis this is reflected in the posterior standard deviations of the timing estimates for the individual parameters: The higher the SNR, i.e. the clearer the transition, the lower the posterior standard deviation. To illustrate this, the Figure below shows the marginal posterior standard deviations of the start point of the transitions as functions of the SNR for the NGRIP record.

[Figure]

As we propagate these uncertainties to the lead/lag calculations and the combined estimates we take this into account. This is also clear when plotting the leads/lags relative to Na as a function of the SNR in the respective parameter which does not show any relationship with the SNR:

[Figure]

We have added this discussion and the two Figures above to the Appendix of the manuscript.

2. Can the authors rule out the influence of water isotope diffusion on the difference in timing of the d18O signals and those of Ca and layer thickness? Based on analysis of NGRIP (Gkinis et al 2014) and a similar site in Antarctica (Jones et al 2017, 2018), I'd guess diffusion lengths are on the order of 5-10cm through this interval and so are not insignificant compared to the annual layer thickness. Such diffusion lengths can have meaningful influence on the inter-annual and even decadal variability in the water isotope record (Jones et al 2018). I imagine that correcting the records used here for the potential influence of diffusion, if that would even be sensible, is far beyond the scope of this study. However, it seems entirely reasonable to estimate the influence (if any) of the smoothing implied by diffusion on the timing identified by the transition model fitting procedure. If you take identical idealized ramps, and smooth one with a time-scale reflective of water isotope diffusion lengths, will the fitting procedure identify the same start, mid, and end points? I suspect these effects, if any, are small, though we are only talking about lags of a few years.

Yes, the smoothing by the isotope diffusion would influence the shape of the transition. It is however difficult to say what the exact effect on the estimated onsets would be as it also increases the SNR by reducing the high-frequency variability that is captured by the noise term.

That being said, in the case of DO onsets, where the d18O signal rapidly increases, any diffusion/smoothing of that signal would yield to a slight shift towards earlier onsets, i.e. would reduce the lag between Ca and d18O.

Nevertheless, this is an interesting and important thing to check. To do so, we examined the auto-correlation time that is estimated by the fitting algorithm. Any diffusion of the signal would lead higher auto-correlation times estimated for d18O than for all the other parameters, even though the AR(1) process is not an ideal description for the autocorrelation introduced through gaussian smoothing.

[Figure]

This Figure shows these estimates for all parameters in NGRIP. In comparison to Ca, d18O exhibits similar auto-correlation lengths throughout the record. Indeed all the records show an increase in auto-correlation length with increasing depth with is a direct consequence of the decrease in the resolution in terms of years due to the layer thinning.

We have added the following to the discussion (P12, L29 ff)

"Due to the small timing differences between the records it is worth noting that water isotope records from polar ice cores are subject to smoothing by diffusion. For the NGRIP isotope record the diffusion length at the end of the last glacial is on the order of five to ten centimeters (Gkinis et al., 2014), influencing the high frequency variability. For the analysis here the diffusion means that the rapid increase of the

δ¹⁸O signal at the D–O onsets would be slightly shifted towards earlier times, leading to lower apparent lags between the aerosol records and the water isotope record. Thus, the inferred lead of Ca over d18O can be regarded as a conservative estimate."

*Gkinis, V. et al. (2014), Water isotope diffusion rates from the NorthGRIP ice core for the last 16,000 years – Glaciological and paleoclimatic implications, Earth and Planetary Science Letters, 405132-141, doi:10.1016/j.epsl.2014.08.022*

3. The most interesting conclusion of this study to my mind is the authors statement that "at face value, this sequence of events suggests that the collapse of North Atlantic sea-ice cover is not the initial trigger for the DO events..." Because of its potential wide interest, this statement deserves some scrutiny. It rests on the authors use of Na as a "qualitative indicator of sea ice cover" in the North Atlantic.

The discussion on Page 5, lines 3-28, highlights the debate over in the interpretation of Na very well. The authors interpretation is laid out on Page 5 lines 20-25. Markle et al 2018 find that most of the millennial variability in Antarctic sea salts can be explained simply by the changes in moisture rainout that are required to explain the water isotope record (these changes also explain most of the changes in Antarctic Ca variability and its relationship to water isotopes in both Antarctica and Greenland (c.f. their Figure 4)). This suggests comparatively small if any changes in sea salt source latitude or strength are needed to explain those Na records (though changes in those things are still possible of course). Is there evidence that this explanation for the sea salts is insufficient in Greenland? Are the observed changes in sea salt for example much larger than what one would expect from temperature dependent rainout alone? It was unclear to me from this discussion that the sea salt source strength (or even mean source latitude) should have a clear relationship to the sea ice edge.

Generally, for Greenland the same mechanism as described by Markle et al. (2018) explains most of the millennial-scale variability as shown in Schüpbach et al. (2018). However, applying the model of Markle et al. (2018) to the DO events would completely preclude any leads or lags between d18O and the aerosols as well between the different aerosols. On shorter timescales dynamical changes in the (co-) transport of aerosols and moisture seem to play a relatively larger role in determining the concentrations observed in the ice cores. Here, this results in the lack of co-evolution between the investigated proxies and can explain the lack of coherence between the records in Markle et al. 2018 at the sub-millennial timescales.

*Schüpbach, S. et al. (2018), Greenland records of aerosol source and atmospheric lifetime changes from the Eemian to the Holocene, Nature Communications, 9, 1-10, doi:10.1038/s41467-018-03924-3.*

If the main way sea ice influences Greenland Na is through its relationship to the variables driving the rainout effect, then a change in sea ice extent doesn't necessarily mean one should have a coincident change in Greenland Na, particularly at interannual timescales.

For example, one can imagine a scenario in which sea ice extent begins to retreat at exactly the same time as the changes observed in Ca initiate. Coincident increases in temperature and moisture removal would cause a decrease in the amount of Na (and Ca) reaching Greenland (as described by the authors). However, if that change in sea ice extent caused an increase in sea salt source production or a northward migration of mean source latitude (both debatable

but reasonable, particularly the latter) this could temporally compensate for the increased removal. This combination of influences could lead to an apparent timing difference in the final Na signal observed in Greenland compared to the actual timing of sea ice changes (an example of the superposition of competing source and rainout factors on polar aerosols is given in the Supplement of Markle et al 2018 c.f. Figures S9). A somewhat similar scenario may be likely for the water isotopes, as changing sea ice extent could drive moisture source effects that could temporally compensate for the decreased depletion driven by simultaneous changes over the ice sheet (these would be of the correct sign to compensate, though it would be hard to assess the potential size of this effect on the isotopes without analyzing deuterium excess records from the same cores). Quantitatively disentangling these influences may be well outside the scope of this study. But this does at least suggest a limitation to using Na as a qualitative indicator of sea ice, and should suggest some commensurate caution in the conclusions drawn based on that interpretation.

Even in the absence of competing influences, uncertainty in the linearity of the sodiumas-sea-ice-extent interpretation poses challenges. Dose a change in the sea ice edge (or extent) of a given size have the same impact on Na in Greenland if the sea ice edge is at 55 degrees North (just for example) versus if the edge is at 65 degrees North? If relationship between changes Greenland Na and the sea ice absolute position is (sufficiently) nonlinear then the changes in sea ice at the start of DO events may not be as detectable in Greenland Na as changes later in the event. This could lead to apparent lags of the Na signal to Ca, even if the change in sea ice initiates at the same time as the change in whatever drives Ca. I'm not sure I think this is particularly likely, but it is certainly plausible. Further, the impact of sea ice on climate isn't linear. Again a given size change in the sea ice edge when the edge is at 55 degrees North (for example) doesn't have the same impact on the surface radiation budget nor the buoyancy forcing of the overturning circulation as when the edge is at 65 degrees North. The relative timing of sodium changes in Greenland don't necessarily rule out sea ice changes as a potential "trigger" of DO events, if climatically meaningful sea ice changes can happen without influencing Greenland Na.

To be sure, there are innumerable, potentially ad hoc, explanations for the data, and it is not the authors responsibility to come up with and then weigh the merits of all such explanations. However the assertion that the lag in sodium seen in Greenland necessitates a lag in changes in sea ice extent with respect to shifts in atmospheric circulation, and that this in turn rules out sea ice changes as the trigger for DO events, is somewhat bold. That this "provides an essential benchmark for climate models" is bolder still. Both of these statements to my mind somewhat outpace the robustness in the interpretation of Na as a qualitative indicator of sea ice extent. I think some slight tempering of the conclusions is merited here and perhaps some discussion of the limitations to the interpretation, or much more compelling evidence is needed that a change in sea ice extent must be seen in the Greenland Na records regardless of other processes. To be clear, I do think the difference in timing of the aerosol species identified here is a compelling target for modeling. And I do think the climatic interpretation offered by the authors is plausible, and one to be taken seriously. It is just not clear to me that these data alone place that strong of constraints on the timing of sea ice changes.

> We agree with the possible complicating factors you pointed out. However, it is difficult to imagine that a "climatically meaningful" sea-ice change can happen without influencing neither Na nor d18O in Greenland ice through the changing source signals, en-route rainout or changes in seasonality or a combination of these factors: Taking the transient DO simulation performed by Vettoretti and Peltier (2015,

2018) as an example, the change in the sea-ice coverage in the North Atlantic results in an immediate increase in moisture cycling (evaporation and precipitation) over the open ocean as well as an increase in temperature in Greenland.
Nevertheless, we agree that barring any direct simulations using both isotope and aerosol models over DO-events, the direct test of our hypothesis will be very difficult. To account for this limitation, we have rephrased the relevant statements in the conclusions to be more careful (P13, L11 ff)

"The progression of environmental changes revealed in the Greenland aerosol records provides a good target for climate models that aim to transiently simulate DO events, preferentially explicitly modeling both water isotope and aerosol transport."

*Vettoretti, G., and Peltier, W. R. (2015), Interhemispheric air temperature phase relationships in the nonlinear Dansgaard-Oeschger oscillation, Geophysical Research Letters, 42,1180-1189, doi:10.1002/2014GL06289*

*Vettoretti, G., and Peltier, W. R. (2018), Fast Physics and Slow Physics in the Nonlinear Dansgaard--Oeschger Relaxation Oscillation, Journal of Climate, 31, 3423-3449, doi:10.1175/JCLI-D-17-0559.1*

Minor points: Page 1, line 8: The authors say "from one of the cores".... Which one?

We have now clarified the sentence in the abstract.

Page 1 line 17: I think that the clause "In the course of the last glacial period" should be moved after the 2nd comma in this line (after "warming episodes,"). This is a possibly silly language thing but: the ice core records reveal that the events were in the last glacial period. It wasn't during the last glacial period that the ice core records revealed these events (that was in the 1980s!).

Done.

Page 1 line 20: This is a language choice the authors should feel free to ignore but "..going along with an almost doubling..." sounds strange to my ear, though I can't tell why. "...coincident with a near-doubling..." sounds better to me.

Adjusted accordingly.

Page 2 line 3: I'd move the "Also" at the start of the sentence to after "Northern Hemisphere".

Done.

Page 2 Lines 20-35: This is a very nice description of several related though distinct ideas. Thanks!

Thank you!

Page 3 lines 15-20: Are the Ca and Na records corrected for the sea-salt vs non sea-salt components of each (e.g. using average Na/Ca mass ratios in average crust and sea salt, as is

common in Antarctic records)? I ask because having looked into this once briefly, it seemed like the corrections often used in Antarctica lead to nonsensical results in Greenland, which was disturbing, and I wasn't sure why.

> No, the records we use here are not "corrected" for the sea-salt and dust contributions. We chose not to do this for two reasons: First and foremost, the Ca/Na ratio in the dust in Greenland is quite different from the crustal average that is typically employed for Antarctic records. The reason for this is the large Ca content of Central Asian Dust. This is also true for Antarctica and has previously been extensively investigated by Bigler et al (2006).
>
> *Bigler, M. et al. (2006), Aerosol deposited in East Antarctica over the last glacial cycle: Detailed apportionment of continental and sea-salt contributions, Journal of Geophysical Research, 11110.1029/2005JD006469.*

Page 6, Line 9: I think there is a missing "of" between "interpretation" and "phase".

> Corrected, thanks.

Page 6, lines 20-21: This is interesting! Nice.

> Thanks.

Page 6, lines 27: "on" should be "one" I think.

> Corrected, thanks.

Page 6, lines 29: If you don't make the assumption that the DO-events have the same imprint in both cores, how would that effect any of your conclusions?

> Making this assumption would allow to combine the estimates from the two cores into one. We chose not to do this to be able to gauge the robustness of our estimates, i.e. whether they agree between the cores.

Page 10, lines 25-30: Might be worth citing Fudge et al 2016 here, they show a strong divergence of accumulation rate and d18O in ice cores at millennial time scales.

> Thank you for pointing out this reference. We added it to the manuscript with the qualification that it deals with Antarctica and not Greenland.

Page 11, lines 3-5: This is nice, convincing analysis.

> Thank you.

Page 11, lines 11: I think this should be "...a coinciding..." Not "...an coinciding..."

> Corrected, thanks.

Page 11, lines 21-22: How big are the additional changes in source need compared to the total variability across a DO event?

To explain the complete amplitude of the stadial/interstadial changes in the Ca concentration in the ice, a source strength change of a factor around 4 is needed. We have added this to the manuscript (P12, L9) and refer the Reviewer to Schüpbach et al. (2018) for further details:

*Schüpbach, S. et al. (2018), Greenland records of aerosol source and atmospheric lifetime changes from the Eemian to the Holocene, Nature Communications, 91-10, doi:10.1038/s41467-018-03924-3.*

Page 11, lines 24-35: This is really nice discussion.

Thank you.

Refs cited: Markle, Bradley R., et al. "Concomitant variability in high-latitude aerosols, water isotopes and the hydrologic cycle." Nature Geoscience (2018): 1.

Jones, Tyler R., et al. "Southern Hemisphere climate variability forced by Northern Hemisphere ice-sheet topography." Nature 554.7692 (2018): 351.

Jones, T. R., et al. "Water isotope diffusion in the WAIS Divide ice core during the Holocene and last glacial." Journal of Geophysical Research: Earth Surface 122.1 (2017): 290-309.

Fudge, T. J., et al. "Variable relationship between accumulation and temperature in West Antarctica for the past 31,000 years." Geophysical Research Letters 43.8 (2016): 3795-3803.

Gkinis, Vasileios, et al. "Water isotope diffusion rates from the NorthGRIP ice core for the last 16,000 years–Glaciological and paleoclimatic implications." Earth and Planetary Science Letters 405 (2014): 132-141.